# Double Check Your State Before Trusting It: Confidence-Aware Bidirectional Offline Model-Based Imagination

**Jiafei Lyu**[1], **Xiu Li**[1,*] **Zongqing Lu**[2*]

[1]Tsinghua Shenzhen International Graduate School, Tsinghua University
[2]School of Computer Science, Peking University
`lvjf20@mails.tsinghua.edu.cn,`
`li.xiu@sz.tsinghua.edu.cn, zongqing.lu@pku.edu.cn`

## Abstract

The learned policy of model-free offline reinforcement learning (RL) methods is often constrained to stay within the support of datasets to avoid possible dangerous out-of-distribution actions or states, making it challenging to handle out-of-support region. Model-based RL methods offer a richer dataset and benefit generalization by generating imaginary trajectories with either trained forward or reverse dynamics model. However, the imagined transitions may be inaccurate, thus downgrading the performance of the underlying offline RL method. In this paper, we propose to augment the offline dataset by using trained bidirectional dynamics models and rollout policies with *double check*. We introduce conservatism by trusting samples that the forward model and backward model agree on. Our method, *confidence-aware bidirectional offline model-based imagination*, generates reliable samples and can be combined with any model-free offline RL method. Experimental results on the D4RL benchmarks demonstrate that our method significantly boosts the performance of existing model-free offline RL algorithms and achieves competitive or better scores against baseline methods.

## 1 Introduction

Offline reinforcement learning (offline RL), also known as batch RL [34], aims at learning from a static dataset that was previously gathered by an unknown behavioral policy. Offline RL is deemed to be promising [16, 14] as online learning requires the agent to continuously interact with the environment, which however may be costly, time-consuming, or even dangerous. The progress in offline RL will undoubtedly scale RL methods to being widely applied in real-world applications, considering the impressive success in computer vision or natural language processing by adopting large-scale offline datasets [9, 6].

Prior off-policy online RL methods [17, 21, 48] are known to fail on fixed offline datasets, even on expert demonstrations [14], due to extrapolation errors [16]. In the offline setting, the agent can overgeneralize from the static dataset, resulting in arbitrarily wrong estimates upon out-of-distribution (OOD) state-action pairs and dangerous action execution. To address this issue, recent model-free offline RL algorithms compel the learned policy to stay close to the behavioral policy [16, 31, 72], or incorporate some penalties into the critic [50, 32, 29]. However, such approaches often suffer from loss of generalization capability [73, 69], since they purposely avoid OOD states or actions.

Model-based offline RL methods, instead, enrich the logged dataset by generating synthetic samples with the trained forward or reverse (backward) dynamics model [26, 75, 69, 74]. These methods

---

[*]Corresponding Authors

36th Conference on Neural Information Processing Systems (NeurIPS 2022).

benefit from better generalization thanks to richer transition samples. Intuitively, the performance of the agent is largely confined by the quality of the model-generated data, i.e., learning on bad states or actions will negatively affect the policy via backpropagation. Unfortunately, there is no guarantee that reliable transitions can be generated by the trained forward or backward dynamics model [2].

In this paper, we aim to generate reliable transitions for offline RL via a *double check mechanism*. The intuition behind this lies in the fact that humans often do double check when they are uncertain and need to be cautious. Besides the forward model, we train the backward model to generate simulated rollouts backward and use one to check whether the synthetic samples the other generated are credible. To be specific, we train bidirectional dynamics models along with bidirectional rollout policies. Instead of injecting pessimism into value estimation, we introduce *conservatism into transition*, i.e., only samples that the forward model and reverse model agree on are trusted.

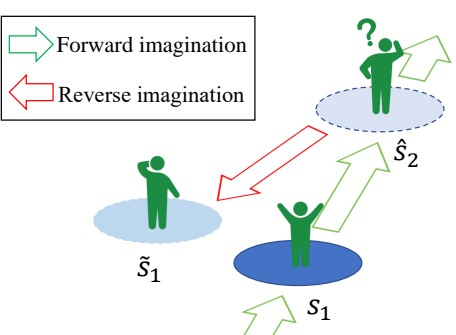

Figure 1: Illustration of the basic idea of confidence-aware bidirectional offline model-based imagination. To ensure that the forward imagination $\hat{s}_2$ from $s_1$ is valid and reasonable, one needs to 'look back' to check whether the imagined previous state $\tilde{s}_1$ based on $\hat{s}_2$ is similar to $s_1$. We trust $\hat{s}_2$ if the deviation between $s_1$ and $\tilde{s}_1$ is small.

We use Figure 1 to further illustrate our insight, where we take forward transition generation as an example, of which the process is identical to the reverse setting. Starting from $s_1$, the forward model predicts next state $\hat{s}_2$. However, it is hard for the agent to decide whether $\hat{s}_2$ is trustworthy. One natural solution, which follows human's way of reasoning [22], is backtracking where it comes from, i.e., looking backward to trace previous state $\tilde{s}_1$, and check whether the imagined state $\tilde{s}_1$ based on $\hat{s}_2$ is different from the true state $s_1$. We are confident to $\hat{s}_2$ if $\tilde{s}_1$ is similar to $s_1$ and vice versa.

To this end, we propose Confidence-Aware Bidirectional Offline Model-Based Imagination (CABI), which is a simple yet effective data augmentation method. CABI generally guarantees the reliability of the generated samples via the double check mechanism, and can be combined with any off-the-shelf model-free offline RL methods, e.g., BCQ [16] and TD3_BC [15], to enjoy better generalization in a conservative manner. Extensive experimental results on the D4RL benchmarks [14] show that CABI significantly boosts the performance of the base model-free offline RL methods, and achieves competitive or even better scores against recent model-free and model-based offline RL methods.

## 2 Related Work

In this paper, we consider offline reinforcement learning [34, 38], which defines the task of learning from a static dataset that was collected by an unknown behavior policy. Applications of offline RL include robotics [45, 57, 54], healthcare [19, 70], recommendation system [59, 64], etc.

**Model-free offline RL.** Since it is risky to execute out-of-support actions, existing offline RL algorithms are often designed to constrain the policy search within the support of the static offline dataset. They realize it via importance sampling [53, 63, 39, 49, 18], explicit or implicit policy constraints [16, 31, 72, 35, 40, 78], learning conservative critics [32, 44, 30, 43, 29, 42], and quantifying estimation uncertainty [73, 76, 10]. Recently, sequential modeling is also explored in the offline RL setting [7, 24, 47]. Despite these advances, model-free offline RL methods suffer from loss of generalization beyond the dataset [69], and CABI is proposed to mitigate it.

**Model-based offline RL.** Model-based offline RL algorithms benefit from better generalization as the static dataset is extended by the synthetic samples generated from the trained forward [56, 13, 1] or reverse dynamics model [69]. These methods heavily rely on uncertainty quantification [52, 75, 26, 11], compelling the policy towards the behavior policy [65, 46], representation learning [36, 54], and penalizing Q-values [74]. However, it is hard to judge whether the transitions generated by the trained dynamics model are reliable, and poor imagined samples will negatively affect the performance of the agent. Recently, [75, 41] explore how inaccurate rollouts that are well penalized can still be useful for model-based training. However, they work only for model-based offline RL methods.

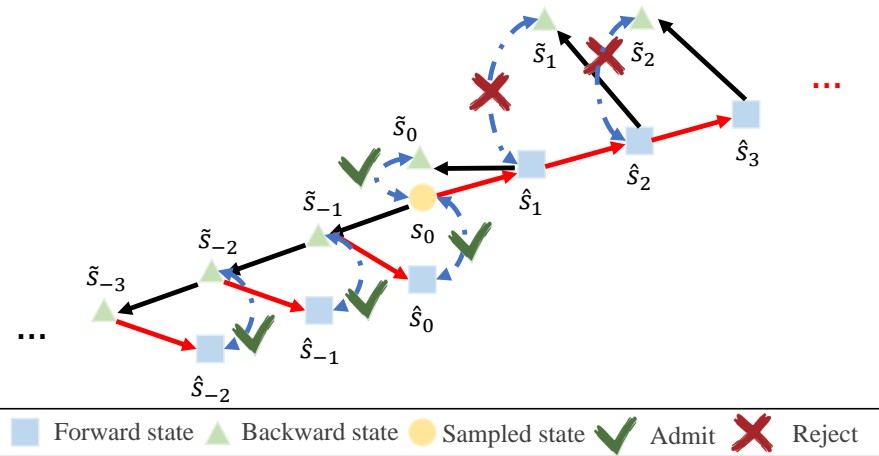

| ■ Forward state | ▲ Backward state | ● Sampled state | ✔ Admit | ✖ Reject |

Figure 2: Illustration of the double check mechanism in bidirectional modeling. We use the trained opposite dynamics model to check whether the synthetic transitions generated by the current dynamics model are reliable. We are confident about the transitions that can be traced back to the starting state with small errors (green check) and reject those with large disagreements (red cross).

There are also some studies that focus on trajectory pruning [26, 77], while they involve uncertainty measurement. CABI, instead, ensures reliable imaginations by conducting double check with the forward and backward models, which fully exploits the advantages of bidirectional modeling.

**Model-based online RL.** Model-based online RL methods achieve superior sample efficiency [61, 25, 4, 23] by learning a dynamics model of the environment and planning with the model [62, 68, 71]. Learning a backward dynamics model that produces traces towards the aimed state is also widely explored [12, 37, 20, 33]. Among them, most similar to our work is [33], which leverages bidirectional model rollouts for reduced compounding error in the online setting. However, the main differences are: (1) CABI is proposed to augment the fixed dataset instead of performing policy optimization in a model-based way; (2) CABI interpolates a double check mechanism for reliable imaginations; (3) Model predictive control (MPC) [5] is not involved in CABI.

## 3 Preliminaries

We study RL under Markov Decision Process (MDP) specified by a tuple $\langle \mathcal{S}, \mathcal{A}, \rho_0, p, r, \gamma \rangle$, where $\mathcal{S}$ is the state space, $\mathcal{A}$ is the action space, $\rho_0$ denotes initial state distribution, $p(s'|s, a)$ is the stochastic transition dynamics, $r(s, a) : \mathcal{S} \times \mathcal{A} \mapsto \mathbb{R}$ is the reward function, and $\gamma \in [0, 1)$ is the discount factor. The policy $\pi(a|s) : \mathcal{S} \times \mathcal{A} \mapsto \mathbb{R}_+$ is a mapping from states to a probability distribution over actions. The goal of RL is to obtain a policy $\pi$ such that the expected discounted cumulative rewards can be maximized, $\max_\pi J_{\rho_0}(\pi) := \mathbb{E}_{s \sim \rho_0, a_t \sim \pi(\cdot|s_t), s_{t+1} \sim p(\cdot|s_t, a_t)} \left[ \sum_{t=0}^\infty \gamma^t r(s_t, a_t) \right]$. In online RL, the agent learns from the experience collected from the interactions with the environment. However, in the offline setting, both interaction and exploration are infeasible, and the agent can only get access to the logged static dataset $\mathcal{D}_{\text{env}} = \{(s, a, r, s')\}$, which was gathered in advance by the unknown behavior policy. Since the fixed dataset $\mathcal{D}_{\text{env}}$ is typically a subset of full space $\mathcal{S} \times \mathcal{A}$, the generalization beyond the raw dataset becomes challenging. Model-based RL mitigates this issue by learning a dynamics model $\hat{p}(s'|s, a)$ and reward function $\hat{r}(s, a)$, and generating synthetic transitions to augment the dataset. However, there is no guarantee that the generated samples are reliable (see Section 4.1), and we focus on addressing this issue in this paper.

## 4 Confidence-Aware Bidirectional Offline Model-Based Imagination

In this section, we first use a toy example to illustrate the necessity of training bidirectional models with the double check mechanism. Then, we give the detailed framework of our method, Confidence-Aware Bidirectional Offline Model-Based Imagination (CABI).

### 4.1 You Need to Double Check Your State

Many model-free offline RL algorithms suffer from poor generalization as they are trained on a fixed dataset with limited samples. Model-based methods extend the logged dataset by generating synthetic transitions from the trained dynamics model. Despite such an advantage, they lack a mechanism for checking the *reliability* of the generated samples. If the model is inaccurate, poor transition samples that lie in the out-of-support region of the dataset can be generated, which may downgrade the performance of offline RL algorithms.

Human beings tend to conduct *double check* when they are uncertain about the outcome, e.g., clinical medicine research [60], autonomous driving [67], etc. Inspired by this nature, we propose to train bidirectional dynamics models and admit the samples where the forward model and backward model have few disagreements instead of roughly trusting all generated samples. In this way, we introduce conservatism into the transition itself instead of the critic or the actor networks. We give the illustration of double check mechanism in Figure 2.

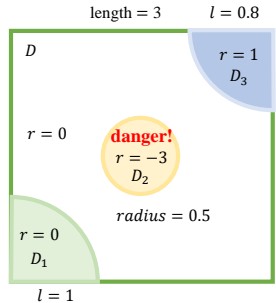

Figure 3: RiskWorld.

We argue that either forward dynamics model or backward dynamics model is unreliable, and bidirectional modeling in conjunction with the double check mechanism is critical for trustworthy sample generation. We verify this by designing a toy task, 2-dimensional environment with continuous state space and action space, namely RiskWorld, as shown in Figure 3. The central point of the square region in RiskWorld is $(0, 0)$, and the state space gives $D := [-1.5, 1.5] \times [-1.5, 1.5]$. Each episode, the agent randomly starts at the region $D_1 := \{(x, y) | (x + 1.5)^2 + (y + 1.5)^2 \leq 1, x < 0, y < 0\}$ and takes actions $a \in [-0.5, 0.5]$. There is a danger zone $D_2 := \{(x, y) | x^2 + y^2 \leq 0.5^2\}$, and the done flag would turn into true if the agent steps into this region, along with a reward of $-3$. The agent will receive a reward of $+1$ if it lies in $D_3 := \{(x, y) | (x - 1.5)^2 + (y - 1.5)^2 \leq 0.8^2, x < 1.5, y < 1.5\}$, and 0 if it locates at $D \backslash (D_2 \cup D_3)$.

We run a random policy in RiskWorld for $10^4$ timesteps to collect an offline dataset. Figure 4(a) shows the state distribution (blue cross) of the dataset, where there are no transitions in $D_2$ (red circle area) as the episode terminates if the agent steps into $D_2$. To compare different ways of imagination, we train a forward model, a backward model, and a bidirectional model with the double check mechanism on this dataset. The training epoch is set to be 100, and the rollout horizon is set to be 3 for all of them. The detailed experimental setup is available in Appendix A.

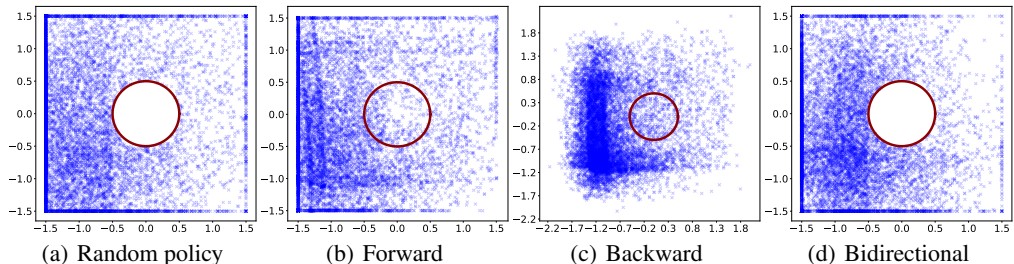

(a) Random policy     (b) Forward     (c) Backward     (d) Bidirectional

Figure 4: Visualizations of the offline dataset collected by a random policy (a), and synthetic transitions generated by a forward model (b), a backward model (c), a bidirectional model with the double check mechanism (d), based on the offline dataset in (a).

We use the trained forward model, reverse model, and bidirectional model to generate $10^4$ transition samples, and plot the state distributions of their generated samples respectively. As shown in Figure 4(b), the forward model generates many samples in the danger zone $D_2$ (red circle area). Figure 4(c) reveals that the reverse model generates a lot of illegal samples that lie out of the state space $D$, and also many transitions that lie in the dangerous area $D_2$. These all show evidence that both the forward and backward dynamics model fail to output reliable transitions. However, we observe in Figure 4(d) that bidirectional modeling with the double check mechanism successfully produces reliable and conservative synthetic samples, i.e., out-of-support or dangerous samples are not included, because

---

**Algorithm 1** CABI (Model Training)

---

**Input:** Dataset $\mathcal{D}_{\text{env}}$, iterations $N_1, N_2$, learning rate $\alpha_\psi, \alpha_\phi, \alpha_\theta, \alpha_\omega$

1: Randomly initialize forward model parameters $\psi$ and reverse model parameters $\phi$
2: **for** $i = 1$ to $N_1$ **do**
3:     Compute $\mathcal{L}_\psi^{\text{fwd}}$ and $\mathcal{L}_\phi^{\text{bwd}}$ via (1) and (2)
4:     Update model parameters: $\psi \leftarrow \psi - \alpha_\psi \nabla_\psi \mathcal{L}_\psi^{\text{fwd}}$, $\phi \leftarrow \phi - \alpha_\phi \nabla_\phi \mathcal{L}_\phi^{\text{bwd}}$
5: **end for**
6: Randomly initialize forward and backward rollout policy parameters $\theta, \omega$
7: **for** $i = 1$ to $N_2$ **do**
8:     Compute $\mathcal{L}_{\text{fvae}}$ and $\mathcal{L}_{\text{bvae}}$ via (3) and (4)
9:     Update forward and reverse rollout policy: $\theta \leftarrow \theta - \alpha_\theta \nabla_\theta \mathcal{L}_{\text{fvae}}$, $\omega \leftarrow \omega - \alpha_\omega \nabla_\omega \mathcal{L}_{\text{bvae}}$
10: **end for**

---

the forward model and backward model have large disagreements at those states. We hence argue that bidirectional modeling with double check is necessary for reliable data generation in offline RL.

### 4.2 Bidirectional Models Learning in CABI

**Bidirectional dynamics models training.** Our bidirectional modeling models transition probability and reward function simultaneously, i.e., the forward model $\hat{p}_\psi(s', r|s, a)$ and the reverse model $\hat{p}_\phi(s, r|s', a)$ parameterized by $\psi$ and $\phi$ respectively. The forward model $\hat{p}_\psi(s', r|s, a)$ represents the probability of the next state and corresponding reward given the current state and action, and the backward model $\hat{p}_\phi(s, r|s', a)$ outputs the probability of the current state and reward using the next state and action as input. We assume that the predicted reward function $\hat{r}(s, a)$ only depends on the current state $s$ and action $a$, then the unified model can be decomposed as $\hat{p}_\psi(s', r|s, a) = \hat{p}(s'|s, a)\hat{p}(r|s, a)$ and $\hat{p}_\phi(s, r|s', a) = \hat{p}(s|s', a)\hat{p}(r|s, a)$. We denote the loss functions for the forward model and backward model as $\mathcal{L}_\psi^{\text{fwd}}$ and $\mathcal{L}_\phi^{\text{bwd}}$ respectively, and optimize them by maximizing the log-likelihood via (1) and (2), where $\mathcal{D}_{\text{env}}$ is the raw static dataset.

$$\mathcal{L}_\psi^{\text{fwd}} = \mathbb{E}_{(s,a,r,s') \sim \mathcal{D}_{\text{env}}} \left[ -\log \hat{p}_\psi(s', r|s, a) \right], \tag{1}$$

$$\mathcal{L}_\phi^{\text{bwd}} = \mathbb{E}_{(s,a,r,s') \sim \mathcal{D}_{\text{env}}} \left[ -\log \hat{p}_\phi(s, r|s', a) \right]. \tag{2}$$

Following prior work [75, 26], we train an ensemble of bootstrapped probabilistic dynamics models, which has been widely demonstrated to be effective in model-based RL [8, 23]. Each model in the ensemble is parameterized by a multi-layer neural network, which outputs a Gaussian distribution $\mathcal{N}(\mu, \Sigma)$. Detailed hyperparameter setup for dynamics models training is deferred to Appendix C.

**Bidirectional rollout policies training.** We additionally train bidirectional generative models, which serve as rollout policies, and are used to generate actions to augment the static dataset. We model the rollout policy with a conditional variational autoencoder (CVAE) [27, 58, 16], which offers diverse actions while staying within the span of the dataset. CVAE is made up of an encoder $E$ that outputs the latent variable $z$ under the Gaussian distribution, and a decoder $D$ that maps $z$ to the desired space. We denote the forward rollout policy as $G_\theta^{\text{fwd}}(s)$ parameterized by $\theta = \{\xi_1, \nu_1\}$ where $\xi_1$ is the parameter of the encoder $E_{\xi_1}^{\text{fwd}}(s, a)$ and $\nu_1$ is the parameter of the decoder $D_{\nu_1}^{\text{fwd}}(s, z)$. The forward rollout policy is then trained by maximizing its variational lower bound, which is equivalent to minimizing the following loss:

$$\mathcal{L}_{\text{fvae}}(\theta) = \mathbb{E}_{(s,a,r,s') \sim \mathcal{D}_{\text{env}}, z \sim E_{\xi_1}^{\text{fwd}}(s,a)} \left[ \left( a - D_{\nu_1}^{\text{fwd}}(s, z) \right)^2 + D_{\text{KL}} \left( E_{\xi_1}^{\text{fwd}}(s, a) \| \mathcal{N}(0, \mathbf{I}) \right) \right], \tag{3}$$

where $D_{\text{KL}}(\cdot\|\cdot)$ denotes the KL-divergence, and $\mathbf{I}$ is an identity matrix. The first term of RHS of (3) represents the reconstruction loss where we want the decoded action to approximate the real action. Then for action generation, we first sample latent vector $z$ from the multivariate Gaussian distribution $\mathcal{N}(0, \mathbf{I})$, and then pass it with the current state $s$ into the decoder $D_{\nu_1}^{\text{fwd}}(s, z)$ to output the action.

Similarly, the backward rollout policy $G_\omega^{\text{bwd}}(s')$ parameterized by $\omega$ contains an encoder $E_{\xi_2}^{\text{bwd}}(s', a)$ and a decoder $D_{\nu_2}^{\text{bwd}}(s', z)$, $\omega = \{\xi_2, \nu_2\}$. The loss function of the backward rollout policy gives:

$$\mathcal{L}_{\text{bvae}}(\omega) = \mathbb{E}_{(s,a,r,s') \sim \mathcal{D}_{\text{env}}, z \sim E_{\xi_2}^{\text{bwd}}(s',a)} \left[ \left( a - D_{\nu_2}^{\text{bwd}}(s', z) \right)^2 + D_{\text{KL}} \left( E_{\xi_2}^{\text{bwd}}(s', a) \| \mathcal{N}(0, \mathbf{I}) \right) \right]. \tag{4}$$

---

**Algorithm 2** CABI (Data Generation)

---

**Input:** Offline dataset $\mathcal{D}_{\mathrm{env}}$, horizon $H$, iteration $N$

1: Initialize model replay buffer $\mathcal{D}_{\mathrm{model}} \leftarrow \emptyset$
2: **for** $i = 1$ to $N$ **do**
3:     Sample state $s_t$ and next state $s_{t+1}$ from $\mathcal{D}_{\mathrm{env}}$
4:     **for** $j = 0$ to $H - 1$ **do**
5:         Obtain forward rollout $\{s_{t+j}, a_{t+j}, r_{t+j}, s_{t+1+j}\}$ from $s_{t+j}$ by drawing samples from the forward model $\hat{p}_\psi$ and forward rollout policy $G_\theta^{\mathrm{fwd}}$
6:         Generate backward state $\tilde{s}_{t+j}$ from $s_{t+1+j}$, and evaluate the deviation of $\tilde{s}_{t+j}$ from $s_{t+j}$
7:         Get backward rollout $\{s_{t-j}, a_{t-j}, r_{t-j}, s_{t+1-j}\}$ from $s_{t+1-j}$ by drawing samples from the backward model $\hat{p}_\phi$ and backward rollout policy $G_\omega^{\mathrm{bwd}}$
8:         Generate forward state $\hat{s}_{t+1-j}$ from $s_{t-j}$. Evaluate the deviation of $\hat{s}_{t+1-j}$ from $s_{t+1-j}$
9:         Add selected imaginations into $\mathcal{D}_{\mathrm{model}}$
10:     **end for**
11: **end for**
12: Get composite dataset $\mathcal{D}_{\mathrm{total}} \leftarrow \mathcal{D}_{\mathrm{env}} \bigcup \mathcal{D}_{\mathrm{model}}$
13: Get the final policy $\pi_{\mathrm{out}}$ with *any* model-free offline RL algorithm based on the dataset $\mathcal{D}_{\mathrm{total}}$

---

We then draw $z$ from the Gaussian distribution $\mathcal{N}(0, \mathbf{I})$, and draw the action from the action decoder $D_{\nu_2}^{\mathrm{bwd}}(s', z)$ with the next state $s'$ and latent variable $z$ as input.

We present the detailed procedure for the model training part of CABI in Algorithm 1.

### 4.3 Conservative Data Augmentation with CABI

After the bidirectional dynamics models and bidirectional rollout policies are well trained, we utilize them to generate imaginary samples. Each time, we sample a state $s_t$ from the raw dataset $\mathcal{D}_{\mathrm{env}}$ to produce imagined forward trajectory $\hat{\tau}^{\mathrm{fwd}} = \{s_{t+j}, a_{t+j}, r_{t+j}, s_{t+1+j}\}_{j=0}^{H-1}$ with the forward dynamics model $\hat{p}_\psi$ and forward rollout policy $G_\theta^{\mathrm{fwd}}$, and sample the next state $s_{t+1}$ from $\mathcal{D}_{\mathrm{env}}$ to generate imagined reverse trajectory $\hat{\tau}^{\mathrm{bwd}} = \{s_{t-j}, a_{t-j}, r_{t-j}, s_{t+1-j}\}_{j=0}^{H-1}$ with the reverse dynamics model $\hat{p}_\phi$ and reverse rollout policy $G_\omega^{\mathrm{bwd}}$. For each step in the rollout horizon $H$, we do double check and reject those badly imagined synthetic transitions.

To be specific, when performing forward imagination from $s_t$ and generating synthetic next state $\hat{s}_{t+1}$, we trace back from $\hat{s}_{t+1}$ with the reverse model, and get the backward state $\tilde{s}_t$. We evaluate the deviation of $\tilde{s}_t$ from $s_t$, and trust $\hat{s}_{t+1}$ if the deviation is small. Similarly, starting from the state $s_{t+1}$, we backtrack its previous state $\tilde{s}_t$ with the backward dynamics model, and then look forward from $\tilde{s}_t$ to get $\hat{s}_{t+1}$ with the forward dynamics model. We trust $\tilde{s}_t$ if the deviation of $\hat{s}_{t+1}$ from $s_{t+1}$ is small.

We keep those trustworthy rollouts and gather them to get the model buffer $\mathcal{D}_{\mathrm{model}}$. We combine the synthetic model buffer $\mathcal{D}_{\mathrm{model}}$ with $\mathcal{D}_{\mathrm{env}}$ to obtain the final buffer $\mathcal{D}_{\mathrm{total}}$, i.e., $\mathcal{D}_{\mathrm{total}} = \mathcal{D}_{\mathrm{env}} \cup \mathcal{D}_{\mathrm{env}}$. We then can train *any* model-free offline RL algorithms based on the composite dataset.

One naïve way for implementing double check mechanism is to set a threshold $\delta$, and admit the transition if $\|s_t - \tilde{s}_t\|_2 \leq \delta$ for forward imagination, or $\|s_{t+1} - \hat{s}_{t+1}\|_2 \leq \delta$ for backward imagination. However, such a method lacks flexibility, and one may need to carefully tune $\delta$ per dataset based on the strong prior knowledge about the dataset, which impedes the application of double check mechanism. We resort to sorting the transitions in a mini-batch by the state deviation from small to large and keep the top $k\%$ of them that have the smallest deviation. We keep 20% transitions that have the smallest deviation for all of our experiments in Section 5 (empirical study on $k$ is available in Appendix C).

Our method is confidence-aware and conservative as we only admit the transitions that the forward model and backward model agree on, thus excluding those poor transitions from the model buffer $\mathcal{D}_{\mathrm{model}}$. The full procedure for the data generation part of CABI is available in Algorithm 2.

Table 1: Normalized average score comparison of CABI+BCQ against different baselines on the Adroit "-v0" tasks, where score 0 represents the performance of a random policy and 100 corresponds to an expert policy performance. The results are averaged over the final 10 evaluations and 5 different random seeds. The highest mean scores are in **bold**.

| Task Name | CABI+BCQ | BCQ | UWAC | BEAR | BC | AWR | CQL | MOPO | COMBO |
|---|---|---|---|---|---|---|---|---|---|
| pen-cloned | 54.7±2.0 | 44.0 | 33.1 | 26.5 | **56.9** | 28.0 | 39.2 | -2.1 | -2.4 |
| pen-human | **75.1**±1.5 | 68.9 | 21.7 | -1.0 | 34.4 | 12.3 | 37.5 | 9.7 | 27.7 |
| pen-expert | **127.6**±2.0 | 114.9 | 111.9 | 105.9 | 85.1 | 111.0 | 107.0 | -0.6 | 11.5 |
| door-cloned | **0.5**±0.2 | 0.0 | 0.0 | -0.1 | -0.1 | 0.0 | 0.4 | -0.1 | 0.0 |
| door-human | 1.7±0.1 | 0.0 | 2.1 | -0.3 | 0.5 | 0.4 | **9.9** | -0.2 | -0.3 |
| door-expert | **105.3**±0.5 | 99.0 | 104.1 | 103.4 | 34.9 | 102.9 | 101.5 | -0.2 | 4.9 |
| relocate-cloned | -0.2±0.0 | -0.3 | -0.3 | -0.3 | **-0.1** | -0.2 | **-0.1** | -0.3 | **-0.1** |
| relocate-human | 0.1±0.1 | **0.5** | **0.5** | -0.3 | 0.0 | 0.0 | 0.2 | -0.3 | -0.3 |
| relocate-expert | **105.9**±1.0 | 41.6 | 105.6 | 98.6 | 101.3 | 91.5 | 95.0 | -0.2 | 17.2 |
| hammer-cloned | **4.3** ±1.6 | 0.4 | 0.4 | 0.3 | 0.8 | 0.4 | 2.1 | 0.2 | 0.4 |
| hammer-human | 3.1±2.2 | 0.5 | 1.1 | 0.3 | 1.5 | 1.2 | **4.4** | 0.2 | 0.2 |
| hammer-expert | **128.9**±0.9 | 107.2 | 110.6 | 127.3 | 125.6 | 39.0 | 86.7 | 0.3 | 0.3 |
| Total Score | **607.0** | 476.7 | 490.8 | 460.3 | 440.8 | 386.5 | 483.8 | 6.4 | 59.1 |

## 5 Experiments

In this section, we combine CABI with off-the-shelf model-free offline RL algorithms and conduct extensive experiments on the D4RL benchmarks [14]. In Section 5.1, we combine CABI with BCQ [16], and evaluate it on the challenging Adroit dataset to show the effectiveness of conservative data augmentation with CABI. We present a detailed ablation study in Section 5.2, where we aim to answer the following questions: (1) Is the double check mechanism a critical component for CABI? (2) How does CABI compare with the forward/reverse imagination? (3) How does CABI compare against other augmentation methods, e.g., random selection? Furthermore, we incorporate CABI with another recent model-free offline RL method, TD3_BC [15], and evaluate it on the MuJoCo datasets, to show the generality and advantages of CABI. We additionally combine CABI with IQL [28] and evaluate the performance of CABI+IQL on both Adroit tasks and MuJoCo tasks. Due to the space limit, the results are deferred to Appendix G.

### 5.1 Performance on Challenging Adroit Dataset

We demonstrate the benefits of CABI by combining it with BCQ and evaluating it on the challenging Adroit dataset [55]. Adroit dataset involves controlling a 24-DoF simulated robotic hand that aims at hammering a nail, opening a door, twirling a pen, or picking/moving a ball. It contains three types of datasets for each task (*human*, *cloned*, and *expert*), yielding a total of 12 datasets. This domain is very challenging for prior methods to learn from because the dataset is made up of narrow human demonstrations on a sparse reward, high-dimensional robotic manipulation task.

We summarize the overall results in Table 1, where we compare CABI+BCQ against recent model-free offline RL methods, such as UWAC [73], CQL [32], BCQ [16], and model-based offline RL methods, such as MOPO [75], and COMBO [74]. We run MOReL and COMBO on these datasets with our reproduced code. Results of MOPO and UWAC on the Adroit domain are acquired by running their official codebases, and the results of the rest baselines are taken directly from [14]. All methods are run over 5 different random seeds and normalized average scores are reported in Table 1. We only report the standard deviation for CABI+BCQ, and the full table is deferred to Appendix H.

As shown, CABI significantly boosts the performance of vanilla BCQ on almost all datasets, achieving a total score of **607.0** vs. 476.7 of BCQ. CABI+BCQ also surpasses the baseline model-free and model-based offline RL methods on 7 out of 12 datasets and achieves the highest total score.

It is worth noting that model-based offline RL methods generally fail on the Adroit tasks, because (1) the dataset distribution is narrow and high-dimensional, making it challenging for the trained forward dynamics model to generate accurate and reliable transitions; (2) the actions in the synthetic transitions are generated by the actor during the training process, thus the error may accumulate if

Table 2: Normalized average score comparison on the Adroit tasks between CABI+BCQ, forward imagination+BCQ, backward imagination+BCQ, and BOMI+BCQ, where $\pm$ captures standard deviation. The results are averaged over the final 10 evaluations and 5 different random seeds. The highest mean scores are in **bold**.

| Task name | BCQ | +Forward | +Backward | +BOMI | +CABI |
|---|---|---|---|---|---|
| pen-cloned | 44.0 | 41.2±1.1 | 36.8±6.6 | 43.4±6.1 | **54.7**±2.0 |
| pen-human | 68.9 | 57.8±9.3 | 60.9±5.6 | 49.6±1.4 | **75.1**±1.5 |
| pen-expert | 114.9 | 114.4±5.4 | 118.5±4.7 | 121.8±1.2 | **127.6**±2.0 |
| door-cloned | 0.0 | 0.0±0.0 | 0.0±0.0 | 0.0±0.0 | **0.5**±0.2 |
| door-human | 0.0 | -0.1±0.1 | 0.0±0.1 | 0.0±0.1 | **1.7**±0.1 |
| door-expert | 99.0 | 104.2±0.3 | 103.7±0.2 | 102.5±1.2 | **105.3**±0.5 |
| relocate-cloned | -0.3 | -0.3±0.0 | -0.3±0.0 | **-0.2**±0.0 | **-0.2**±0.0 |
| relocate-human | **0.5** | 0.0±0.0 | 0.0±0.0 | 0.0±0.1 | 0.1±0.1 |
| relocate-expert | 41.6 | 72.9±2.0 | 76.8±6.8 | 80.1±9.3 | **105.9**±1.0 |
| hammer-cloned | 0.4 | 1.7±0.1 | 0.4±0.1 | 3.1±3.8 | **4.3**±1.6 |
| hammer-human | 0.5 | 2.0±0.2 | 2.8±0.5 | 2.1±0.7 | **3.1**±2.2 |
| hammer-expert | 107.2 | 126.8±1.0 | 126.9±1.0 | 126.8±1.3 | **128.9**±0.9 |
| Total score | 476.7 | 520.6 | 526.5 | 529.2 | **607.0** |

the actor is updated towards a wrong direction. CABI, instead, alleviates the underlying issues via adopting the CVAE for action generation and conducting double check on state prediction.

## 5.2 Ablation Study

**Is the double check mechanism critical?** To answer this question, we exclude the double check mechanism in CABI and admit all generated synthetic samples from bidirectional models, which gives rise to Bidirectional Offline Model-based Imagination (BOMI). We evaluate CABI+BCQ and BOMI+BCQ on the Adroit tasks with identical parameter configuration over 5 different random seeds and show the average normalized scores in Table 2. It can be seen that BOMI brings some performance improvement on most of the tasks via data augmentation with bidirectional models and rollout policies. However, the generated data may be unreliable (we observe a performance drop in *pen-cloned, pen-human*), which impedes the benefits of bidirectional data augmentation. Such concern can be alleviated with the aid of the double check mechanism. As illustrated in Table 2, CABI+BCQ outperforms BOMI+BCQ on most tasks and incurs a much better total score.

**CABI against forward/backward imagination.** We incorporate BCQ with the pure forward model, backward model, and CABI, and conduct extensive experiments on the Adroit tasks over 5 different random seeds. The forward model and reverse model are trained with the same configuration as CABI. The results are summarized in Table 2. It can be seen that either the forward or reverse model results in limited improvement, which is consistent with the results of BOMI. As previously discussed, the forward model and reverse model may generate unreliable transitions. We see such evidence as the performance of BCQ falls on some of the tasks (e.g., *pen-cloned*) if trained on mere forward or reverse imagination. BCQ+CABI, instead, outperforms BCQ+Forward and BCQ+Backward on all tasks. Hence, we conclude that CABI guarantees trustworthy transitions for training, and brings improvement on almost all of the tasks.

**CABI against other augmentation methods.** We further compare CABI against three data augmentation methods: (1) CABI-random where we replace the CVAE with the random policy as the rollout policy in CABI; (2) R-20 where we *randomly* select 20% synthetic transitions for bidirectional imagination; (3) EV-20 where we select 20% transitions with the smallest *ensemble variance* for bidirectional imagination, i.e., we evaluate the variance of the output of the ensemble of the forward and backward dynamics models and reject those with large variance. We use BCQ as the base algorithm and run experiments on four Adroit tasks for these augmentation methods with identical parameter setup as CABI (e.g., real data ratio). The results in Table 4 show that CABI performs consistently better than these methods. Since the data augmentation process of CABI is isolated from the policy optimization, we cannot leverage a random rollout policy because the generated actions of a random policy may possibly lie out of the span of the dataset, which can negatively affect the performance of the agent. Hence, CVAE is critical to ensure a safe and conservative data

Table 3: Normalized average score comparison of CABI+TD3_BC vs. baseline methods on the D4RL MuJoCo "-v0" dataset, where score 0 corresponds to a random policy performance and 100 corresponds to an expert policy performance. The results are averaged over the final 10 evaluations and 5 different random seeds. The highest mean scores are in **bold**.

| Task Name | CABI+TD3_BC | TD3_BC | UWAC | MOPO | BCQ | BC | CQL | FisherBRC |
|---|---|---|---|---|---|---|---|---|
| halfcheetah-random | 15.1±0.4 | 10.2 | 2.3 | **35.4** | 2.2 | 2.0 | 21.7 | 32.2 |
| hopper-random | **11.9**±0.1 | 11.0 | 9.8 | 11.7 | 10.6 | 9.5 | 10.7 | 11.4 |
| walker2d-random | 6.4±1.5 | 1.4 | 3.8 | **13.6** | 4.9 | 1.2 | 2.7 | 0.6 |
| halfcheetah-medium-replay | 44.4±0.2 | 43.3 | 38.9 | **53.1** | 38.2 | 34.7 | 41.9 | 43.3 |
| hopper-medium-replay | 31.3±0.7 | 31.4 | 18.0 | **67.5** | 33.1 | 19.7 | 28.6 | 35.6 |
| walker2d-medium-replay | 29.4±1.3 | 25.2 | 8.4 | 39.0 | 15.0 | 8.3 | 15.8 | **42.6** |
| halfcheetah-medium | **45.1**±0.1 | 42.8 | 37.4 | 42.3 | 40.7 | 36.6 | 37.2 | 41.3 |
| hopper-medium | **100.4**±0.3 | 99.5 | 30.3 | 28.0 | 54.5 | 30.0 | 44.2 | 99.4 |
| walker2d-medium | **82.0**±0.4 | 79.7 | 17.4 | 17.8 | 53.1 | 11.4 | 57.5 | 79.5 |
| halfcheetah-medium-expert | **105.0**±0.2 | 97.9 | 40.6 | 63.3 | 64.7 | 67.6 | 27.1 | 96.1 |
| hopper-medium-expert | **112.7**±0.0 | 112.2 | 95.4 | 23.7 | 110.9 | 89.6 | 111.4 | 90.6 |
| walker2d-medium-expert | **108.4**±1.3 | 101.1 | 14.8 | 44.6 | 57.5 | 12.0 | 68.1 | 103.6 |
| halfcheetah-expert | **107.6**±0.9 | 105.7 | 104.0 | - | 89.9 | 105.2 | 82.4 | 106.8 |
| hopper-expert | **112.4**±0.1 | 112.2 | 109.1 | - | 107.0 | 111.5 | 111.2 | 112.3 |
| walker2d-expert | **108.6**±1.5 | 105.7 | 88.4 | - | 102.3 | 56.0 | 103.8 | 79.9 |
| Total Score | **1020.7** | 979.3 | 618.6 | - | 784.6 | 595.3 | 764.3 | 974.6 |

augmentation. Meanwhile, relying on the ensemble variance for data selection is not trustworthy as the models in the ensemble are trained on the identical data and may all incur wrong predictions but small variance.

Table 4: Normalized average score comparison on four Adroit tasks. The results are averaged over the final 10 evaluations and 5 different random seeds. CABI-random denotes the rollout policy in CABI is a random policy. R-20 denotes **R**andomly keep 20% transitions, EV-20 denotes keep 20% samples that have the smallest **E**nsemble **V**ariance.

| Task Name | BCQ | +CABI | +R-20 | +EV-20 | +CABI-random |
|---|---|---|---|---|---|
| pen-cloned | 44.0 | **54.7**±2.0 | 41.2±3.0 | 40.4±2.0 | 37.6±7.8 |
| pen-expert | 114.9 | **127.6**±2.0 | 112.6±5.6 | 118.8±2.5 | 106.3±3.7 |
| hammer-cloned | 0.4 | **4.3**±1.6 | 0.9±0.6 | 0.4±0.1 | 0.3±0.0 |
| hammer-expert | 107.2 | **128.9**±0.9 | 104.2±24.6 | 125.5±5.5 | 103.8±1.5 |

## 5.3 Broad Results on MuJoCo Dataset

To show the generality of our method, we integrate CABI with another recent model-free offline RL method, TD3_BC [15], and conduct experiments on 15 MuJoCo datasets. We widely compare CABI+TD3_BC against other recent model-free offline RL methods, such as FisherBRC [29], UWAC [73], CQL [32], and model-based batch RL method, MOPO [75]. We run CABI+TD3_BC over 5 different random seeds. We also run UWAC using the official codebase on the MuJoCo datasets over 5 different random seeds. The results of TD3_BC, BC, CQL, FisherBRC are taken directly from [15], and the results of other baseline methods are taken from [73].

The experimental results in Table 3 reveal that our approach exceeds all baseline methods on 10 out of 15 datasets, and is the strongest in terms of the total score. On almost all of the tasks, we observe performance improvement with CABI over the base TD3_BC algorithm. Unfortunately, with the existence of behavioral cloning term, the performance improvement upon TD3_BC is limited. Still, the experimental results in Table 1 and 3 show that CABI is a powerful data augmentation method and can boost the performance of the base model-free offline RL methods.

# 6 Conclusion and Limitations

In this paper, we follow human nature and propose to do *double check* during synthetic transition generation to ensure that the imagined samples are conservative and accurate. We admit samples that the forward model and reverse model agree on. Our method, CABI, involves training bidirectional dynamics models and rollout policies and can be combined with *any* off-the-shelf model-free offline RL algorithms. Extensive experiments on the D4RL benchmarks show that our method significantly boosts the performance of the base model-free offline RL method, and can achieve competitive or better performance against recent baseline methods. For future work, it is interesting to evaluate CABI in the online setting and investigate whether it can benefit model-based online RL as well.

The major limitation of our proposed method lies in the computation cost as we train bidirectional dynamics models and rollout policies. However, since CABI is isolated from policy optimization, we can enhance the dataset beforehand.

## Acknowledgments and Disclosure of Funding

This work was supported in part by the Science and Technology Innovation 2030-Key Project under Grant 2021ZD0201404, in part by the NSF China under Grant 61872009. The authors would like to thank the anonymous reviewers for their valuable comments and advice.

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
