# A Experimental Setup of Toy RiskWorld Task

In this section, we give the detailed experimental setup of our toy RiskWorld task. RiskWorld is a 2-dimensional, continuous state space, continuous action space environment as shown in Figure 3. We suppose the central point of RiskWorld is $(0,0)$, and the permitted range of RiskWorld gives $D := [-1.5, 1.5] \times [-1.5, 1.5]$, i.e., the length of RiskWorld is 3. The state information in RiskWorld is composed of the coordinates of the agent, i.e., $s = (x, y), x, y \in [-1.5, 1.5]$. There is a dangerous area $D_2$ locating at the central point with radius 0.5, i.e., $D_2 := \{(x, y)|x^2 + y^2 \leq 0.5^2\}$. The agent randomly starts at a point in $D_1 := \{(x, y)|(x + 1.5)^2 + (y + 1.5)^2 \leq 1, x < 0, y < 0\}$, and can take actions $a \in [-0.5, 0.5]$. There is also a high reward zone locating at $D_3 := \{(x, y)|(x - 1.5)^2 + (y - 1.5)^2 \leq 0.8^2, x < 1.5, y < 1.5\}$. The reward function $r(s, a)$ is defined in (5).

$$r(s, a) = \begin{cases} -3, & \text{if } s \in D_2 := \{(x, y)|x^2 + y^2 \leq 0.5^2\}, \\ 1, & \text{if } s \in D_3 := \{(x, y)|(x - 1.5)^2 + (y - 1.5)^2 \leq 0.8^2, x < 1.5, y < 1.5\}, \\ 0, & \text{if } s \in D \backslash (D_2 \cup D_3). \end{cases} \quad (5)$$

The agent will receive a large minus reward of $-3$ if it steps into the dangerous zone $D_2$, and the done flag will turn into *true*. The agent is not allowed to step out of the legal region $D$. The episode length for RiskWorld is set to be 300. The RiskWorld is intrinsically a sparse reward environment. We run a random policy on RiskWorld for $10^4$ timesteps and log the transition data it collected during interactions to form a static dataset *RiskWorld-random*.

Model-based reinforcement learning (RL) learns either forward dynamics or reverse dynamics of the environment [62, 20], and can produce imaginary transitions for training, which has been widely demonstrated to be effective in improving the sample efficiency of RL in the online setting [4, 23]. The forward dynamics model $\hat{p}_\psi(s', r|s, a)$ predicts the next state and the corresponding reward function given the current state and action, and the reverse dynamics model $\hat{p}_\phi(s, r|s', a)$ outputs the previous state and reward signal given action and the next state. Bidirectional modeling combines both forward dynamics model and backward dynamics model.

To compare different ways of imagination, i.e., the forward imagination, reverse imagination, and bidirectional imagination with the double check mechanism, we train a forward dynamics model, a backward dynamics model, and a bidirectional dynamics model on RiskWorld-random dataset, respectively. We represent the forward and reverse dynamics model by training a probabilistic neural network. The forward model $\hat{p}_\psi(s', r|s, a)$ parameterized by $\psi$ receives the current state and action as input, and outputs a multivariate Gaussian distribution that predicts the next state and reward as shown in (6).

$$\hat{p}_\psi(s', r|s, a) = \mathcal{N}(\mu_\psi(s, a), \boldsymbol{\Sigma}_\psi(s, a)), \quad (6)$$

where $\mu_\psi$ and $\boldsymbol{\Sigma}_\psi$ represent the mean and variance of the forward model $\hat{p}_\psi(s', r|s, a)$, respectively.

Similarly, for the backward model $\hat{p}_\phi(s, a|s', a)$ parameterized by $\phi$, it adopts the next state and action as input and outputs a multivariate normal distribution predicting reward signal and the previous state (see (7)).

$$\hat{p}_\phi(s, a|s', a) = \mathcal{N}(\mu_\phi(s', a), \boldsymbol{\Sigma}_\phi(s', a)), \quad (7)$$

where $\mu_\phi$ and $\boldsymbol{\Sigma}_\phi$ denote the mean and variance of the backward model $\hat{p}_\phi(s, r|s', a)$, respectively.

The probabilistic neural network is modeled by a multi-layer neural network that consists of 4 feedforward layers with 400 hidden units. We adopt *swish* activation for each intermediate layer. Following prior works [23, 75, 26], we train an ensemble of seven such probability neural networks for both the forward and backward model. We use a hold-out set made up of 1000 transitions to validate the performance of the trained dynamics, and select the five models that have the best performance accordingly. When performing forward or reverse imagination, we randomly pick one model out of the five best model candidates to generate synthetic trajectories per step. Considering the simplicity of the toy RiskWorld task, we train both the forward dynamics model and reverse dynamics model for 100 epochs, and the rollout length (horizon) is set to be 3 for the forward model, reverse model, and bidirectional model. We use the trained dynamics model (forward, reverse, bidirectional) to generate $10^4$ imaginary transition samples, and log their model buffer respectively. We plot in Figure 4 the model buffer of these dynamics models and the raw static dataset obtained by running the random policy.

# B Datasets and Evaluation Setting on the D4RL Benchmarks

In this section, we give a detailed description of the datasets we used in this paper, and also describe the evaluation setting that is adopted on the D4RL benchmarks [14]. D4RL is specially designed for evaluating offline RL (or batch RL) algorithms, which covers the dimensions that offline RL may encounter in practical applications, such as passively logged data, human demonstrations, etc.

## B.1 Adroit datasets and MuJoCo datasets

The Adroit dataset involves controlling a 24-DoF simulated Shadow Hand robot to perform tasks like hammering a nail, opening a door, twirling a pen, and picking/moving a ball, as shown in Figure 5. The Adroit domain is super challenging for even online RL algorithms because: (1) the dataset contains narrow human demonstrations; (2) this domain solves sparse reward, high-dimensional robotic manipulation tasks. There are four tasks in the dataset, and there are three types of datasets for each task, *cloned, human, expert*. **human:** a small number of demonstrations operated by a human (25 trajectories per task). **expert:** a large amount of expert data from a fine-tuned RL policy. **cloned:** a large amount of data generated by performing imitation learning on the human demonstrations, running the policy, and mixing the data at a 50-50 ratio with the demonstrations. Dataset mixing is involved for *cloned* as the cloned policies themselves fail on the tasks, making the dataset otherwise hard to learn from.

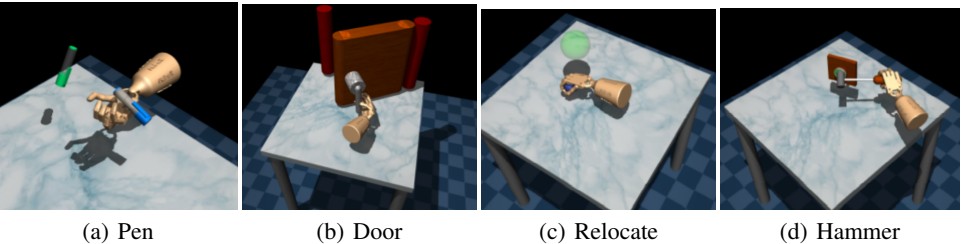

| (a) Pen | (b) Door | (c) Relocate | (d) Hammer |

Figure 5: Adroit datasets. There are four tasks, namely pen, door, relocate, and hammer.

The MuJoCo dataset is collected during the interactions with the continuous action environments in Gym [3] simulated by MuJoCo [66]. We adopt three tasks in this dataset, *halfcheetah, hopper, walker2d* as illustrated in Figure 6. Each task in the MuJoCo dataset contains five types of datasets, *random, medium, medium-replay, medium-expert, expert*. **random:** a large amount of data from a random policy. **medium:** experiences collected from an early-stopped SAC policy for 1M steps. **medium-replay:** replay buffer of a policy trained up to the performance of the medium agent. **expert:** a large amount of data gathered by the SAC policy that is trained to completion. **medium-expert:** a large amount of data by mixing the medium data and expert data at a 50-50 ratio.

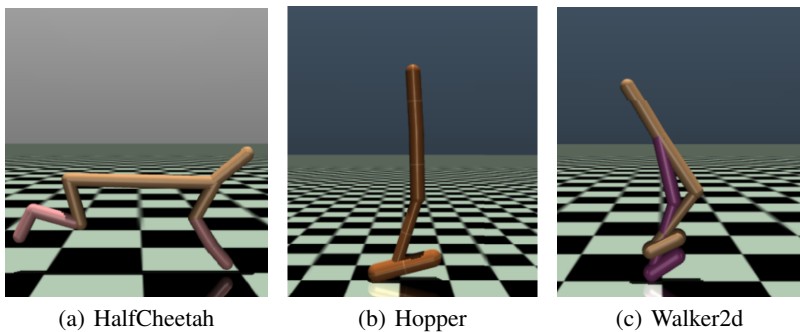

| (a) HalfCheetah | (b) Hopper | (c) Walker2d |

Figure 6: MuJoCo datasets. We conduct experiments on halfcheetah, hopper, and walker2d tasks.

Note that the Adroit dataset is qualitatively different from the MuJoCo Gym dataset because (1) there do not exist human demonstrations in the MuJoCo dataset; (2) the reward is dense in MuJoCo, making it less challenging to learn from; (3) the dimension of transitions in MuJoCo is low compared with Adroit. It is hard for even online RL algorithms to learn useful policies on the Adroit tasks, while it is easy for online RL methods to achieve superior performance on the MuJoCo environments.

## B.2 Evaluation setting in D4RL

D4RL suggests using the normalized score metric to evaluate the performance of the offline RL algorithms [14]. Denote the expected return of a random policy on the dataset as $C_r$ (reference min score), and the expected return of an expert policy as $C_e$ (reference max score). Suppose that an offline RL algorithm achieves an expected

return of $C$ after training on the given dataset. Then the normalized score $\tilde{C}$ is given by (8).

$$\tilde{C} = \frac{C - C_r}{C_e - C_r} \times 100 = \frac{C - \text{performance of random policy}}{\text{performance of expert policy} - \text{performance of random policy}} \times 100. \quad (8)$$

The normalized score ranges roughly from 0 to 100, where 0 corresponds to the performance of a random policy and 100 corresponds to the performance of an expert policy. We give the detailed reference min score $C_r$ and reference max score $C_e$ in Table 5, where all of the tasks share the same reference min score and reference max score across different types of datasets.

Table 5: The referenced min score and max score for the Adroit dataset and MuJoCo dataset in D4RL.

| Domain | Task Name | Reference min score $C_r$ | Reference max score $C_e$ |
|--------|-----------|---------------------------|---------------------------|
| Adroit | pen | 96.26 | 3076.83 |
| Adroit | door | $-56.51$ | 2880.57 |
| Adroit | relocate | $-6.43$ | 4233.88 |
| Adroit | hammer | $-274.86$ | 12794.13 |
| MuJoCo | halfcheetah | $-280.18$ | 12135.0 |
| MuJoCo | hopper | $-20.27$ | 3234.3 |
| MuJoCo | Walker2d | 1.63 | 4592.3 |

## C  Implementation Details and Hyperparameters

### C.1  Implementation details

In this section, we give implementation details and hyperparameters for Confidence-Aware Bidirectional Offline Model-Based Imagination (CABI). We represent the approximated forward dynamics and reward model, and backward dynamics and reward model by training a probabilistic neural network. The configuration of the probabilistic neural network is identical to Appendix A. That is, the forward model and reverse model are modeled as a multivariate Gaussian distribution with mean $\mu$ and variance $\Sigma$. For the forward model $\hat{p}_\psi(s', r|s, a)$ parameterized by $\psi$, it accepts the current state and action as input and generates the next state and reward. The backward model $\hat{p}_\phi(s, r|s', a)$ parameterized by $\phi$ receives the next state and current action as input and outputs the former state and scalar reward. The probabilistic neural network is modeled by a multi-layer neural network that contains 4 feedforward layers, with 400 hidden units in each layer, and a *swish* activation in each intermediate layer. We train an ensemble of seven such probabilistic neural networks and select the best five models based on their performance on a hold-out set made up of 1000 transitions from the offline dataset.

As a data augmentation method, CABI does not actively generate actions during the training process, i.e., the policy optimization process is isolated from the data generation process. We use a conditional variational autoencoder (CVAE) to approximate the behavior policy in the static dataset. We give brief introduction to the VAE in Appendix E. The CVAE $G$ contains an encoder $E$ and a decoder $D$. Both the encoder and the decoder in the forward rollout policy and backward rollout policy contain two intermediate layers with 750 hidden units each layer. We adopt *relu* activation for each intermediate layer. Specifically, we train a forward rollout policy with a CVAE $G_\theta^{\text{fwd}}(s)$, which contains an encoder $E_{\xi_1}(s, a)$ and a decoder $D_{\nu_1}(s, z)$, $\theta = \{\xi_1, \nu_1\}$, and a reverse rollout policy $G_\omega^{\text{bwd}}(s') = \{E_{\xi_2}(s', a), D_{\nu_2}(s', z)\}$, where $\omega = \{\xi_2, \nu_2\}$. Note that the forward rollout policy $G_\theta^{\text{fwd}}(s)$ and reverse rollout policy $G_\omega^{\text{bwd}}(s')$ sample actions using stochastic inference from an underlying latent space, so as to increase diversity in the generated actions.

Intrinsically, CABI can be combined with *any* off-the-shelf model-free offline RL algorithms. In this work, we incorporate CABI with BCQ [16] and TD3_BC [15], and conduct extensive experiments on the Adroit dataset and MuJoCo dataset on the D4RL benchmarks, respectively. There are generally three steps when combining CABI with model-free offline RL methods: (1) **Model training.** We first train bidirectional dynamics models and bidirectional rollout policies using the raw static offline dataset $\mathcal{D}_{\text{env}}$; (2) **Data generation.** After the bidirectional models and rollout policies are well trained, we utilize them to generate imaginary trajectories, while conducting double check and admitting high-confidence transitions simultaneously. This will induce the model generated dataset $\mathcal{D}_{\text{model}}$; (3) **Policy optimization.** We then merge the real dataset $\mathcal{D}_{\text{env}}$ with the imagined dataset $\mathcal{D}_{\text{model}}$ to form a composite dataset $\mathcal{D}_{\text{total}}$, i.e., $\mathcal{D}_{\text{total}} = \mathcal{D}_{\text{env}} \cup \mathcal{D}_{\text{model}}$. The mini-batch samples used for training the model-free offline RL algorithms come from $\mathcal{D}_{\text{env}}$ and $\mathcal{D}_{\text{model}}$. To be specific, we define the ratio of data come from $\mathcal{D}_{\text{env}}$ (real data) as $\eta$. Suppose we use a mini-batch size of $N$ for training the algorithm. Then for each optimization step, we sample $\eta N$ samples from $\mathcal{D}_{\text{env}}$, and sample $(1 - \eta)N$ samples from $\mathcal{D}_{\text{model}}$. We pick the optimal real data ratio $\eta$ among $\{0.1, 0.3, 0.5, 0.7, 0.9\}$ using grid search.

**Influence of** $k$**.** For the double check mechanism, we keep the top 20% of samples in a mini-batch that the forward model and backward model have the smallest disagreements. As explained in Section 4.3, we do not set a threshold and then admit samples that the deviation of the forward model and backward model on them are smaller than the threshold, because it requires human knowledge and the threshold is task-specific. However, in real-world problems, we cannot always have full knowledge about the system. For better flexibility and generality, we choose to keep top $k$% samples. Note that if we use $k = 0$, then the influence of data augmentation will be excluded. If we use $k = 100$, then CABI will degenerate into Bidirectional Model-based Offline Imagination (BOMI), where there is no double check procedure. Intuitively, if $k$ is small, few samples can be left, which may negatively affect the advantages of data augmentation with a model. While if $k$ is large, some poorly imagined transitions may be included in the model buffer. We simply set $k = 20$ by default, and use it throughout our experiments on the Adroit dataset and MuJoCo dataset. We conduct experiments on two types of MuJoCo datasets, *random, medium*, with varied $k$ in $\{0, 10, 20, 50, 100\}$ over 5 different random seeds. The results are shown in Table 6, where we observe performance drop for both large $k$ and small $k$. We hence set $k = 20$ for all of our experiments.

Table 6: Normalized average score of CABI+TD3_BC on two datasets, random and medium, from MuJoCo dataset with different values of $k$. CABI+TD3_BC with $k = 0$ degenerates into vanilla TD3_BC, and CABI+TD3_BC with $k = 100$ turns into BOMI+TD3_BC. We **bold** the highest mean.

| Task Name | $k = 0$ | $k = 10$ | $k = 20$ | $k = 50$ | $k = 100$ |
|---|---|---|---|---|---|
| halfcheetah-random | 10.2 | 14.8 | **15.1** | 14.0 | 11.4 |
| hopper-random | 11.0 | 10.3 | **11.9** | 10.6 | 9.3 |
| walker2d-random | 1.4 | 4.9 | **6.4** | 5.8 | 4.3 |
| halfcheetah-medium | 42.8 | 44.6 | **45.1** | 44.4 | 44.3 |
| hopper-medium | 99.5 | 99.7 | **100.4** | 32.3 | 3.1 |
| walker2d-medium | 79.7 | 78.7 | **82.0** | 79.8 | 78.6 |

**Computation time and compute infrastructure.** The computation time for CABI ranges from 4 to 14 hours on the MuJoCo and Adroit tasks. The model training time differs on different types of datasets, e.g., it takes much less time to train our bidirectional models and rollout policies on *medium-replay* and *human* datasets (about 40 minutes), while it takes comparatively longer time to train on other types of datasets (about 2-6 hours). TD3_BC consumes about 3 hours to run on all MuJoCo datasets, and BCQ takes about 6-8 hours to train on the Adroit tasks. We run both CABI+BCQ and CABI+TD3_BC for $1 \times 10^6$ timesteps. We additionally run IQL [28], and the results are presented in Section G. IQL takes about 3-7 hours to run on all tasks with $1 \times 10^6$ timesteps. We give detailed compute infrastructure in Appendix F.

**Discussion on CABI and ROMI [69].** A recent work, Reverse Offline Model-based Imagination (ROMI) [69], explores the data augmentation in offline RL via training a reverse dynamics model. It is worth noting that we do not directly compare with ROMI+BCQ as there are *many secondary components* in the codebase of ROMI (https://github.com/wenzhe-li/romi), e.g., prioritized experience replay, modifying state information, adopting varied rollout policies for different domains, etc. CABI represents the rollout policy using only conditional variational autoencoder (CVAE). Also, ROMI assumes that the termination functions are known. However, CABI does not include any prior knowledge about termination conditions, even on simple MuJoCo tasks. That generally follows the key claims in the recent work of [15]. For a fair comparison, we disable the forward model as well as the double check mechanism in CABI to get our reverse model. As for the comparison with forward imagination, we disable the double check mechanism and the reverse dynamics part of CABI to get our pure forward model.

**Ensemble variance.** In the main text, we compare our CABI against EV-20, which rejects transitions with large ensemble variance. The ensemble variance is calculated in the following way where we take the forward imagination as an example: suppose we have an ensemble of forward dynamics models $f_i(\cdot|s, a), i = 1, \ldots, N$ with the ensemble size $N$. Then each model in the ensemble predicts a next state $s_i'$. We then have a collection of next state $\{s_i'\}, i = 1, \ldots, N$. We then randomly pick one next state while recording the variance in the ensemble at the same time. We reject the generated next state if the variance in the ensemble is large. That is, we evaluate the variance of $\{s_i'\}, i = 1, \ldots, N$. We sort the transitions in a batch by their calculated variance, and only trust the top 20% that have the smallest ensemble variance.

**Other implementation details.** On the Adroit tasks, we combine CABI with BCQ, and compare against recent state-of-the-art methods, including vanilla BCQ [16], UWAC [73], CQL [32], MOPO [75], and COMBO [74], etc. On the MuJoCo domain, we incorporate CABI with TD3_BC [15], and compare with baseline methods like FisherBRC [29], UWAC, CQL, BEAR [31], MOPO, etc. Note that we omit some baseline methods, such as AWAC [51] and BRAC [72] in the MuJoCo domain, as they do not obtain good enough performance for comparison. We run UWAC with the official codebase (https://github.com/apple/ml-uwac), and so is MOPO (https://github.com/tianheyu927/mopo). We run MOPO on the Adroit tasks as those results are not reported

in the original paper, and we take the results of MOPO on the MuJoCo datasets from [73] directly. We re-run UWAC on the Adroit domain and MuJoCo domain because, unfortunately, we cannot reproduce the results reported in its original paper. All baseline methods are run for $1 \times 10^6$ timesteps over 5 different random seeds.

## C.2 Hyperparameters

In this subsection, we give the detailed hyperparameter setup for our experiments in Table 7. We keep the top 20% samples in a mini-batch for all tasks. For simplicity, we use identical rollout length for the forward model and backward model. For all of the MuJoCo tasks and most of the Adroit tasks, the rollout length for both the forward model and backward model is set to be 3, which yields a total horizon of 6. On datasets that the model disagreement are comparatively large for long horizons (e.g., *pen-expert*, see Table 9), we set the forward and backward horizon as 1, which leads to a total horizon of 2. We use the forward and backward horizon 5 for *hammer-human* as we experimentally find that it performs better. On other tasks, the forward and backward horizon is set to be 3 by default. Note that the model disagreement on MuJoCo datasets are smaller than 0.1 for all horizons, and the trained forward and backward model well fits MuJoCo datasets. We therefore adopt the forward and backward horizon of 3 for all of these tasks.

Table 7: Hyperparameters setup in our experiments with CABI on the Adroit dataset and MuJoCo dataset. ForH = Forward Horizon, BackH = Backward Horizon.

| Domain | Dataset Type | Task Name | ForH | BackH | Real Data Ratio $\eta$ |
|--------|-------------|-----------|------|-------|------------------------|
| Adroit | human | pen | 1 | 1 | 0.7 (BCQ), 0.5 (IQL) |
| Adroit | human | door | 3 | 3 | 0.7 (BCQ), 0.5 (IQL) |
| Adroit | human | relocate | 3 | 3 | 0.9 (BCQ), 0.5 (IQL) |
| Adroit | human | hammer | 5 | 5 | 0.5 (BCQ), 0.7 (IQL) |
| Adroit | cloned | pen | 1 | 1 | 0.5 (BCQ), 0.5 (IQL) |
| Adroit | cloned | door | 3 | 3 | 0.5 (BCQ), 0.7 (IQL) |
| Adroit | cloned | relocate | 3 | 3 | 0.3 (BCQ), 0.5 (IQL) |
| Adroit | cloned | hammer | 3 | 3 | 0.5 (BCQ), 0.7 (IQL) |
| Adroit | expert | pen | 1 | 1 | 0.7 (BCQ), 0.9 (IQL) |
| Adroit | expert | door | 3 | 3 | 0.7 (BCQ), 0.9 (IQL) |
| Adroit | expert | relocate | 3 | 3 | 0.9 (BCQ), 0.7 (IQL) |
| Adroit | expert | hammer | 1 | 1 | 0.9 (BCQ), 0.5 (IQL) |
| MuJoCo | random | halfcheetah | 3 | 3 | 0.7 (TD3_BC), 0.7 (IQL) |
| MuJoCo | random | hopper | 3 | 3 | 0.1 (TD3_BC), 0.7 (IQL) |
| MuJoCo | random | walker2d | 3 | 3 | 0.1 (TD3_BC), 0.7 (IQL) |
| MuJoCo | medium | halfcheetah | 3 | 3 | 0.7 (TD3_BC), 0.7 (IQL) |
| MuJoCo | medium | hopper | 3 | 3 | 0.9 (TD3_BC), 0.7 (IQL) |
| MuJoCo | medium | walker2d | 3 | 3 | 0.7 (TD3_BC), 0.9 (IQL) |
| MuJoCo | medium-replay | halfcheetah | 3 | 3 | 0.5 (TD3_BC), 0.7 (IQL) |
| MuJoCo | medium-replay | hopper | 3 | 3 | 0.7 (TD3_BC), 0.7 (IQL) |
| MuJoCo | medium-replay | walker2d | 3 | 3 | 0.5 (TD3_BC), 0.9 (IQL) |
| MuJoCo | medium-expert | halfcheetah | 3 | 3 | 0.7 (TD3_BC), 0.9 (IQL) |
| MuJoCo | medium-expert | hopper | 3 | 3 | 0.9 (TD3_BC), 0.9 (IQL) |
| MuJoCo | medium-expert | walker2d | 3 | 3 | 0.7 (TD3_BC), 0.9 (IQL) |
| MuJoCo | expert | halfcheetah | 3 | 3 | 0.7 (TD3_BC), 0.9 (IQL) |
| MuJoCo | expert | hopper | 3 | 3 | 0.9 (TD3_BC), 0.9 (IQL) |
| MuJoCo | expert | walker2d | 3 | 3 | 0.7 (TD3_BC), 0.9 (IQL) |

We search for the best $\eta$ over $\{0.1, 0.3, 0.5, 0.7, 0.9\}$. We find that the real data ratio $\eta = 0.7$ and $\eta = 0.9$ are generally effective for CABI. The best ratio $\eta$ strongly depends on the dataset and may need to be tuned manually. For example, *random* dataset in the MuJoCo domain and *cloned* dataset in the Adroit domain are poor for training naturally, and small $\eta$ is therefore needed. While for *expert* dataset or *medium* dataset, a comparatively large $\eta$ is better.

## D Model Prediction Error and Model Disagreement

In this section, we are interested in exploring (1) can CABI generate more trustworthy transitions in complex environments (2) the model disagreement of the forward and backward models in CABI under different horizons,

Table 8: Comparison of one-step model prediction error of the forward model, reverse model, and bidirectional model with the double check mechanism on the Adroit tasks.

| Task Name | Unidirectional | | Bidirectional (CABI) | |
|---|---|---|---|---|
| | $\epsilon_{\text{fwd}}$ | $\epsilon_{\text{bwd}}$ | $\epsilon_{\text{fwd}}$ | $\epsilon_{\text{bwd}}$ |
| pen-cloned | 837.5 | 777.4 | **751.5** | **603.0** |
| pen-human | 195.0 | 177.8 | **107.5** | **97.8** |
| pen-expert | 169.01 | 179.8 | **143.58** | **149.8** |
| door-cloned | 24.7 | 27.7 | **0.05** | **0.01** |
| door-human | 18.2 | 20.2 | **4.4** | **6.0** |
| door-expert | 4.3 | 10.5 | **1.8** | **6.3** |
| relocate-cloned | 351.9 | 1271.4 | **0.0** | **0.9** |
| relocate-human | 229.5 | 267.4 | **178.6** | **205.1** |
| relocate-expert | 201.5 | 48.3 | **167.5** | **37.9** |
| hammer-cloned | 1330.8 | 1984.3 | **72.3** | **1602.2** |
| hammer-human | 577.9 | 596.4 | **480.9** | **477.8** |
| hammer-expert | 601.1 | 561.4 | **557.2** | **503.4** |

aiming at checking whether the model disagrees with each other more with the increment of the horizon. To begin with, we define the one-step model prediction error to check whether CABI admits more accurate transitions.

**Definition D.1** (Model Prediction Error). Given the static offline dataset $\mathcal{D}_{\text{env}}$, we define one-step model prediction error for forward model $\epsilon_{\text{fwd}}$ and reverse model $\epsilon_{\text{bwd}}$ as:

$$\epsilon_{\text{fwd}} = \mathbb{E}_{\substack{(s,a,r,s')\sim\mathcal{D}_{\text{env}} \\ \hat{s},\hat{r}\sim\hat{p}_\psi(\cdot|s,a)}} \left[ \|s' - \hat{s}\|_2^2 + (r - \hat{r})^2 \right],$$

$$\epsilon_{\text{bwd}} = \mathbb{E}_{\substack{(s,a,r,s')\sim\mathcal{D}_{\text{env}} \\ \tilde{s},\tilde{r}\sim\hat{p}_\phi(\cdot|s',a)}} \left[ \|s - \tilde{s}\|_2^2 + (r - \tilde{r})^2 \right].$$

$\epsilon_{\text{fwd}}$ and $\epsilon_{\text{bwd}}$ generally capture the accuracy of the trained dynamics models, i.e., smaller $\epsilon_{\text{fwd}}$ and $\epsilon_{\text{bwd}}$ indicate better forward and backward dynamics model fitting. Intuitively, the one-step model prediction error of admitted samples in CABI should be smaller than that of the mere forward dynamics model or reverse dynamics model, as only transitions that the forward model and backward model are all confident about are admitted. We verify this by comparing the one-step model error in the forward model, backward model, and CABI, where we keep the top 20% imagined samples for CABI. The results are presented in Table 8, where we observe CABI leads to significant error drop for both forward and reverse models on all of the tasks. For example, the forward error in *door-cloned* drops from 24.7 to **0.05** and the backward error drops from 27.7 to **0.01**, which reveals that CABI can select reliable and conservative imaginations that well fit the dataset for training.

We then define the model disagreement of forward model and backward model in the following.

**Definition D.2** (Bidirectional Model Disagreement). For a sampled current state $s$ and reward $r$ from a given static offline dataset $\mathcal{D}$, a series of forward states $\hat{s}_i$ and reward signals $\hat{r}_i$ can be generated by utilizing the forward model, $i = 1, \ldots, H$. Denote the imagined backward state and reward based on $\hat{s}_i$ as $\tilde{s}_{i-1}$ and $\tilde{r}_{i-1}$, respectively, $i = 1, \ldots, H$. Then the forward model disagreement is defined as:

$$\epsilon_i^{\text{fwd}} = \begin{cases} \mathbb{E}\left[ \|\tilde{s}_0 - s\|_2^2 + (\tilde{r}_0 - r)^2 \right], & \text{if } i = 1, \\ \mathbb{E}\left[ \|\tilde{s}_{i-1} - \hat{s}_{i-1}\|_2^2 + (\tilde{r}_{i-1} - \hat{r}_{i-1})^2 \right], & \text{if } i \geq 2. \end{cases} \quad (9)$$

Similarly, for a sampled next state $s'$ and reward $r$ from the offline dataset, an imaginary trajectory $\hat{\tau}_{\text{bwd}}$ containing the backward states $\tilde{s}_{-i}$ and rewards $\tilde{r}_{-i}$, $i = 1, \ldots, H$, can be generated with the aid of the backward dynamics model. For each imagined state $\tilde{s}_{-i}$ in $\hat{\tau}_{\text{bwd}}$, its previous state $\hat{s}_{-i+1}$ and reward $\hat{r}_{-i+1}$ are generated by the forward model, $i = 1, \ldots, H$. Then the backward model disagreement is defined as:

$$\epsilon_i^{\text{bwd}} = \begin{cases} \mathbb{E}\left[ \|\hat{s}_0 - s'\|_2^2 + (\hat{r}_0 - r)^2 \right], & \text{if } i = 1, \\ \mathbb{E}\left[ \|\tilde{s}_{-i+1} - \hat{s}_{-i+1}\|_2^2 + (\tilde{r}_{-i+1} - \hat{r}_{-i+1})^2 \right], & \text{if } i \geq 2. \end{cases} \quad (10)$$

**Remark:** The above definition generally capture the disagreement between the forward model and the backward model. Note that the model disagreement is different from the model prediction error defined above even if the rollout length is set as 1. The model prediction error measure how well the forward or backward model fits the transition data, while the model disagreement measures how the forward model and backward model disagree on the transition. We take the forward setting as an example. The forward model prediction error is the deviation between the forward imagined state and reward against the real next state and reward signal, while the forward model disagreement is the deviation between the real *current state* and scalar reward with the backward imagined *current state* and reward based on the forward imagination.

Table 9 details model disagreement comparison of CABI against CABI without double check mechanism, which turns into BOMI, i.e., bidirectional modeling without double check, under different horizons. We perform experiments on 12 Adroit tasks and the sampled mini-batch size is set to be $5 \times 10^4$. As demonstrated in the table, the model disagreement of CABI is significantly smaller than that of BOMI under different rollout steps. It is worth noting that the model disagreement for both CABI and BOMI is irrelevant to the rollout length. The model disagreement generally is small when performing one-step model rollout, and increases if longer horizon imaginations are generated (some datasets like *door-human* are exceptions). We observe that the model disagreement in CABI is much more controllable than BOMI, e.g., on some expert datasets.

Table 9: The model disagreement comparison of **CABI** and **BOMI** under different rollout length. The superscript fwd denotes forward, and bwd denotes backward. The subscript denotes the imagined horizon, e.g., $\epsilon_1^{\text{fwd}}$ represents the forward model disagreement under horizon 1. The best results are in **bold** (smaller is better).

| Task Name | $\epsilon_1^{\text{fwd}}$ | | $\epsilon_2^{\text{fwd}}$ | | $\epsilon_3^{\text{fwd}}$ | | $\epsilon_1^{\text{bwd}}$ | | $\epsilon_2^{\text{bwd}}$ | | $\epsilon_3^{\text{bwd}}$ | |
|---|---|---|---|---|---|---|---|---|---|---|---|---|
| | CABI | BOMI | CABI | BOMI | CABI | BOMI | CABI | BOMI | CABI | BOMI | CABI | BOMI |
| pen-human | **0.23** | 1829.21 | **20.63** | 1608.66 | **21.24** | 1608.94 | **0.23** | 1857.14 | **19.93** | 1693.70 | **19.25** | 1686.21 |
| door-human | **0.53** | 36.03 | **0.21** | 28.06 | **0.21** | 29.04 | **0.53** | 36.19 | **0.40** | 29.38 | **0.38** | 30.07 |
| relocate-human | **0.00** | 268.37 | **5.32** | 237.03 | **5.45** | 236.64 | **0.00** | 265.34 | **4.68** | 193.56 | **4.72** | 195.08 |
| hammer-human | **0.67** | 401.58 | **5.56** | 261.43 | **5.53** | 259.86 | **0.69** | 402.67 | **5.63** | 260.93 | **5.43** | 258.86 |
| pen-cloned | **159.58** | 12048.06 | **373.10** | 22506.36 | **359.56** | 22440.26 | **136.89** | 11624.98 | **359.91** | 22350.11 | **331.18** | 21939.83 |
| door-cloned | **0.00** | 46.35 | **0.00** | 63.75 | **0.00** | 62.89 | **0.00** | 30.68 | **0.00** | 61.03 | **0.00** | 62.96 |
| relocate-cloned | **0.00** | 745.77 | **0.00** | 938.51 | **0.00** | 929.95 | **0.00** | 269.66 | **0.00** | 909.64 | **0.00** | 889.79 |
| hammer-cloned | **0.0** | 3791.10 | **0.01** | 4509.45 | **0.01** | 4395.46 | **0.0** | 1021.85 | **0.02** | 4443.90 | **0.03** | 4447.14 |
| pen-expert | **0.42** | 1965.76 | **21.92** | 9642.54 | **26.05** | 9691.13 | **0.41** | 1953.54 | **56.12** | 9569.12 | **52.69** | 9399.45 |
| door-expert | **0.95** | 257.49 | **0.94** | 261.84 | **1.04** | 263.52 | **1.01** | 258.51 | **0.79** | 276.60 | **0.69** | 276.12 |
| relocate-expert | **0.05** | 649.59 | **1.47** | 770.99 | **2.18** | 784.79 | **0.06** | 652.63 | **0.06** | 631.83 | **0.05** | 631.74 |
| hammer-expert | **1.03** | 6135.13 | **58.07** | 82422.41 | **60.78** | 82633.47 | **1.02** | 6198.06 | **11.24** | 92882.87 | **10.70** | 92102.51 |

The double check mechanism we introduced in the main text selects trustworthy synthetic samples based on the deviation between states, i.e., transition samples with small state deviation will be kept in the model buffer. While we can also trust the transition samples via the model disagreement, i.e., keep transition samples with small model disagreement. We experimentally find that evaluating the deviation between states brings almost the same performance as evaluating the model disagreement under the identical hyperparameter setup. We choose to select transitions according to the deviation between states alone as shown in Algorithm 2 for both space and time saving during data generation process of CABI.

# E  Omitted Background for VAE

In this section, we provide a brief introduction to the variational autoencoder (VAE) [27]. Given a dataset $X = \left\{x^{(i)}\right\}_{i=1}^N$, the VAE is trained to generate samples that come from the same distribution as the data points. That is to say, the goal of a VAE is to maximize $p_\theta(X) = \prod_{i=1}^N p_\theta\left(x^{(i)}\right)$, where $\theta$ is the parameter of the approximate maximum-likelihood (ML) or maximum a posterior (MAP) estimation. To reach this goal, a latent variable $z$ sampled from its posterior distribution $p(z)$ is introduced, and we model a decoder $p_\theta(X|z)$ parameterized by $\theta$. However, directly optimizing the marginal likelihood $p_\theta(X) = \int p(z)p_\theta(X|z)dz$ is intractable. Instead, VAE approximates the true posterior $p_\theta(z|X)$ via training an encoder $q_\phi(z|X)$, and we resort to optimizing the evidence lower bound (ELBO) on the log-likelihood of the data as shown in (11).

$$\max_{\theta,\phi} \log p_\theta(X) \geq \max_{\theta,\phi} \mathbb{E}_{q_\phi(z|X)}[\log p_\theta(X|z)] - D_{\text{KL}}(q_\phi(z|X)\|p_\theta(z)). \tag{11}$$

The first term in the right-hand-side of (11) denotes the reconstruction loss, where $z$ is sampled from $q_\phi(z|X)$. The second term represents the KL-divergence between the learned encoder of $z$ and its true prior. The encoder $q_\phi(z|X)$ is usually set to be a multivariate Gaussian distribution with mean $\mu_\phi$ and variance $\Sigma_\phi$. The prior of the latent variable $z$ is set to be a standard multivariate Gaussian distribution. Optimizing the lower bound in (11) enables the trained model to generate samples similar to the data distribution. After the VAE is well trained, we sample $z$ from the encoder $q_\phi(z|X)$ and pass it through the decoder $p_\theta(X|z)$ to obtain samples.

In this work, we use the conditional variational autoencoder (CVAE) [16] to model the behavior policy in the dataset. CVAE is a variant of the vanilla VAE, which aims to model $p_\theta(X|Y)$. Similar to the original ELBO of VAE, CVAE optimizes the conditional lower bound as shown in (12).

$$\max_{\theta,\phi} \log p_\theta(X|Y) \geq \max_{\theta,\phi} \mathbb{E}_{q_\phi(z|X,Y)}[\log p_\theta(X|z,Y)] - D_{\text{KL}}(q_\phi(z|X,Y)\|p_\theta(z|Y)). \tag{12}$$

# F   Compute Infrastructure

In Table 10, we list the compute infrastructure that we use to run all of the baseline algorithms and experiments.

Table 10: Compute infrastructure.

| CPU | GPU | Memory |
|---|---|---|
| AMD EPYC 7452 | RTX3090×8 | 288GB |

# G   Experimental Results of CABI+IQL

In this section, we additionally incorporate CABI with a recently proposed offline RL method, IQL [28]. IQL learns without querying OOD samples. Such a learning paradigm ensures that the whole learning process is conducted under the support of the dataset, and a safe policy can be learned. However, as we explained in the main text, the datasets often cannot contain all possible transitions. Hence, the generalization capability of IQL is actually limited. With the aid of CABI, such concern can be mitigated to some extent. We conduct experiments on 12 Adroit datasets and 15 MuJoCo datasets over 5 different random seeds. For IQL, we use its official codebase (https://github.com/ikostrikov/implicit_q_learning) to run on all 27 datasets over 5 random seeds with the hyperparameters suggested by the authors. We incorporate CABI with IQL and run CABI+IQL over 5 different random seeds. The forward and backward horizons for CABI+IQL are identical to CABI+BCQ on Adroit tasks and CABI+TD3_BC on MuJoCo datasets. We summarize the results in Table 11 and Table 12.

As shown, CABI boosts the performance of IQL on all 27 datasets of Adroit and MuJoCo. CABI+IQL outperforms baseline methods on 10 out of 12 datasets. While on MuJoCo datasets, CABI+IQL only surpasses baseline methods on 5 out of 15 datasets, due to the fact that the base method IQL itself has poor performance on "-v0" datasets. Nevertheless, CABI+IQL has a total score of 604.1 on Adroit, surpassing the total score 562.5 of the vanilla IQL. CABI+IQL achieves a total score of 909.3 on MuJoCo datasets, while vanilla IQL only has a total score of 860.7. We want to emphasize here that we do not aim to beat the most recent strong baseline methods in this paper, the key point we want to carry here is the conservative data augmentation with CABI is effective and beneficial for the performance improvement over the base offline RL algorithms. The empirical experiments work as the evidence to validate our claim.

# H   Omitted Full Comparison of CABI against Baselines

In this section, we provide the full comparison of CABI against baseline methods as we omit standard deviation for baselines in the main text due to space limitation. We show in Table 13 the full performance comparison of CABI+BCQ against BCQ [16], UWAC [73], CQL [32], MOPO [75], COMBO [74], etc. We additionally compare against SAC. As MOPO and COMBO do not report the performance on the Adroit dataset in their original paper, we run COMBO on the Adroit tasks over 5 different random seeds with our reproduced code, and run MOPO and UWAC with their official codebases on the Adroit tasks over 5 different random seeds, respectively. The results of the rest of the baselines are taken directly from [14].

Table 14 gives the full comparison of CABI+TD3_BC against TD3_BC [15], BCQ [16], UWAC [73], FisherBRC [29], CQL [32], MOPO [75], etc. We additionally compare CABI+BCQ against BEAR here. The results of BC, TD3_BC, CQL, and FisherBRC are taken directly from [15], and the results of UWAC are acquired by running the official codebase over 5 different random seeds. The results of BEAR are taken directly from [73]. We do not report standard deviation for BEAR and BCQ.

Table 11: Normalized average score comparison of CABI+IQL against different baselines on the Adroit "-v0" tasks, where score 0 represents the performance of a random policy and 100 corresponds to an expert policy performance. The highest mean scores are in **bold**.

| Task Name | CABI+IQL | IQL | UWAC | BEAR | BC | AWR | CQL | MOPO | COMBO |
|---|---|---|---|---|---|---|---|---|---|
| pen-cloned | 42.2±6.1 | 35.2±7.3 | 33.1 | 26.5 | **56.9** | 28.0 | 39.2 | -2.1 | -2.4 |
| pen-human | **72.0**±9.1 | 68.7±8.6 | 21.7 | -1.0 | 34.4 | 12.3 | 37.5 | 9.7 | 27.7 |
| pen-expert | **129.1**±0.6 | 118.4±6.9 | 111.9 | 105.9 | 85.1 | 111.0 | 107.0 | -0.6 | 11.5 |
| door-cloned | **0.8**±0.4 | 0.7±0.5 | 0.0 | -0.1 | -0.1 | 0.0 | 0.4 | -0.1 | 0.0 |
| door-human | **11.5**±3.6 | 3.3±1.3 | 2.1 | -0.3 | 0.5 | 0.4 | 9.9 | -0.2 | -0.3 |
| door-expert | **105.7**±0.2 | 105.2±0.3 | 104.1 | 103.4 | 34.9 | 102.9 | 101.5 | -0.2 | 4.9 |
| relocate-cloned | **-0.1**±0.0 | -0.2±0.0 | -0.3 | -0.3 | **-0.1** | -0.2 | **-0.1** | -0.3 | **-0.1** |
| relocate-human | 0.4±0.3 | 0.0±0.0 | **0.5** | -0.3 | 0.0 | 0.0 | 0.2 | -0.3 | -0.3 |
| relocate-expert | **107.4**±0.2 | 105.6±0.5 | 105.6 | 98.6 | 101.3 | 91.5 | 95.0 | -0.2 | 17.2 |
| hammer-cloned | **2.4** ±0.2 | 1.6±1.0 | 0.4 | 0.3 | 0.8 | 0.4 | 2.1 | 0.2 | 0.4 |
| hammer-human | **4.8**±1.8 | 2.3±0.6 | 1.1 | 0.3 | 1.5 | 1.2 | 4.4 | 0.2 | 0.2 |
| hammer-expert | **127.9**±1.2 | 121.7±1.3 | 110.6 | 127.3 | 125.6 | 39.0 | 86.7 | 0.3 | 0.3 |
| Total Score | **604.1** | 562.5 | 490.8 | 460.3 | 440.8 | 386.5 | 483.8 | 6.4 | 59.1 |

Table 12: Normalized average score comparison of CABI+IQL vs. baseline methods on the D4RL MuJoCo "-v0" dataset, where score 0 corresponds to a random policy performance and 100 corresponds to an expert policy performance. The highest mean scores are in **bold**.

| Task Name | CABI+IQL | IQL | UWAC | MOPO | BCQ | BC | CQL | FisherBRC |
|---|---|---|---|---|---|---|---|---|
| halfcheetah-random | 18.4±0.9 | 16.2±0.2 | 2.3 | **35.4** | 2.2 | 2.0 | 21.7 | 32.2 |
| hopper-random | 11.4±0.1 | 9.3±1.8 | 9.8 | **11.7** | 10.6 | 9.5 | 10.7 | 11.4 |
| walker2d-random | 8.0±0.5 | 6.2±2.2 | 3.8 | **13.6** | 4.9 | 1.2 | 2.7 | 0.6 |
| halfcheetah-medium-replay | 42.2±0.2 | 40.5±0.4 | 38.9 | **53.1** | 38.2 | 34.7 | 41.9 | 43.3 |
| hopper-medium-replay | 36.8±0.4 | 33.4±1.1 | 18.0 | **67.5** | 33.1 | 19.7 | 28.6 | 35.6 |
| walker2d-medium-replay | 17.2±0.8 | 15.8±1.7 | 8.4 | 39.0 | 15.0 | 8.3 | 15.8 | **42.6** |
| halfcheetah-medium | 41.6±0.1 | 41.2±0.1 | 37.4 | **42.3** | 40.7 | 36.6 | 37.2 | 41.3 |
| hopper-medium | 40.0±12.9 | 30.7±0.0 | 30.3 | 28.0 | 54.5 | 30.0 | 44.2 | **99.4** |
| walker2d-medium | 55.1±2.3 | 50.8±7.7 | 17.4 | 17.8 | 53.1 | 11.4 | 57.5 | **79.5** |
| halfcheetah-medium-expert | **96.7**±1.3 | 89.0±0.7 | 40.6 | 63.3 | 64.7 | 67.6 | 27.1 | 96.1 |
| hopper-medium-expert | **112.8**±0.2 | 111.5±1.0 | 95.4 | 23.7 | 110.9 | 89.6 | 111.4 | 90.6 |
| walker2d-medium-expert | **104.8**±1.0 | 99.7±2.9 | 14.8 | 44.6 | 57.5 | 12.0 | 68.1 | 103.6 |
| halfcheetah-expert | 104.7±0.9 | 100.8±3.7 | 104.0 | - | 89.9 | 105.2 | 82.4 | **106.8** |
| hopper-expert | **112.8**±0.2 | 112.0±0.0 | 109.1 | - | 107.0 | 111.5 | 111.2 | 112.3 |
| walker2d-expert | **106.8**±3.7 | 103.6±2.0 | 88.4 | - | 102.3 | 56.0 | 103.8 | 79.9 |
| Total Score | 909.3 | 860.7 | 618.6 | - | 784.6 | 595.3 | 764.3 | **974.6** |

Table 13: Normalized average score comparison of CABI+BCQ against different baselines on the Adroit tasks, where score 0 represents the performance of a random policy and 100 corresponds to an expert policy performance. The highest mean scores are in **bold**.

| Task Name | CABI+BCQ | BCQ | UWAC | BEAR | BC | AWR | CQL | SAC-off | MOPO | COMBO |
|---|---|---|---|---|---|---|---|---|---|---|
| pen-cloned | 54.7±2.0 | 44.0 | 33.1±10.1 | 26.5 | **56.9** | 28.0 | 39.2 | 23.5 | -2.1±4.0 | -2.4±1.1 |
| pen-human | **75.1**±1.5 | 68.9 | 21.7±2.2 | -1.0 | 34.4 | 12.3 | 37.5 | 6.3 | 9.7±4.1 | 27.7±10.7 |
| pen-expert | **127.6**±2.0 | 114.9 | 111.9±1.2 | 105.9 | 85.1 | 111.0 | 107.0 | 6.1 | -0.6±1.9 | 11.5±2.3 |
| door-cloned | **0.5**±0.2 | 0.0 | 0.0±0.0 | -0.1 | -0.1 | 0.0 | 0.4 | 0.0 | -0.1±0.1 | 0.0±0.0 |
| door-human | 1.7±0.1 | -0.0 | 2.1±1.6 | -0.3 | 0.5 | 0.4 | **9.9** | 3.9 | -0.2±0.1 | -0.3±0.2 |
| door-expert | **105.3**±0.5 | 99.0 | 104.1±1.8 | 103.4 | 34.9 | 102.9 | 101.5 | 7.5 | -0.2±0.1 | 4.9±1.2 |
| relocate-cloned | -0.2±0.0 | -0.3 | -0.3±0.0 | -0.3 | **-0.1** | -0.2 | **-0.1** | -0.2 | -0.3±0.1 | **-0.1**±0.1 |
| relocate-human | 0.1±0.1 | **0.5** | **0.5**±0.6 | -0.3 | 0.0 | 0.0 | 0.2 | 0.0 | -0.3±0.1 | -0.3±0.1 |
| relocate-expert | **105.9**±1.0 | 41.6 | 105.6±1.6 | 98.6 | 101.3 | 91.5 | 95.0 | -0.3 | -0.2±0.0 | 17.2±3.1 |
| hammer-cloned | **4.3**±1.6 | 0.4 | 0.4±0.0 | 0.3 | 0.8 | 0.4 | 2.1 | 0.2 | 0.2±0.0 | 0.4±0.2 |
| hammer-human | 3.1±2.2 | 0.5 | 1.1±0.6 | 0.3 | 1.5 | 1.2 | **4.4** | 0.5 | 0.2±0.0 | 0.2±0.0 |
| hammer-expert | **128.9**±0.9 | 107.2 | 110.6±20.7 | 127.3 | 125.6 | 39.0 | 86.7 | 25.2 | 0.3±0.0 | 0.3±0.1 |
| Total Score | **607.0** | 476.7 | 490.8 | 460.3 | 440.8 | 386.5 | 483.8 | 72.7 | 6.4 | 59.1 |

Table 14: Normalized average score of CABI+TD3_BC vs. prior state-of-the-art methods on the D4RL MuJoCo dataset, where score 0 corresponds to a random policy performance and 100 corresponds to an expert policy performance.

| Task Name | CABI+TD3_BC | TD3_BC | UWAC | MOPO | BEAR | BCQ | BC | CQL | FisherBRC |
|---|---|---|---|---|---|---|---|---|---|
| halfcheetah-random | 15.1±0.4 | 10.2±1.3 | 2.3±0.0 | **35.4**±2.5 | 25.1 | 2.2 | 2.0±0.1 | 21.7±0.9 | 32.2±2.2 |
| hopper-random | **11.9**±0.1 | 11.0±0.1 | 9.8±0.1 | 11.7±0.4 | 11.4 | 10.6 | 9.5±0.1 | 10.7±0.1 | 11.4±0.2 |
| walker2d-random | 6.4±1.5 | 1.4±0.6 | 3.8±1.3 | **13.6**±2.6 | 7.3 | 4.9 | 1.2±0.2 | 2.7±1.2 | 0.6±0.6 |
| halfcheetah-med-replay | 44.4±0.2 | 43.3±0.5 | 38.9±0.3 | **53.1**±2.0 | 38.6 | 38.2 | 34.7±1.8 | 41.9±1.1 | 43.3±0.9 |
| hopper-med-replay | 31.3±0.7 | 31.4±3.0 | 18.0±5.6 | **67.5**±24.7 | 33.7 | 33.1 | 19.7±5.9 | 28.6±0.9 | 35.6±2.5 |
| walker2d-med-replay | 29.4±1.3 | 25.2±5.1 | 8.4±1.2 | 39.0±9.6 | 19.2 | 15.0 | 8.3±1.5 | 15.8±2.6 | **42.6**±7.0 |
| halfcheetah-medium | **45.1**±0.1 | 42.8±0.3 | 37.4±0.2 | 42.3±1.6 | 41.7 | 40.7 | 36.6±0.6 | 37.2±0.3 | 41.3±0.5 |
| hopper-medium | **100.4**±0.3 | 99.5±1.0 | 30.3±0.3 | 28.0±12.4 | 52.1 | 54.5 | 30.0±0.5 | 44.2±10.8 | 99.4±0.4 |
| walker2d-medium | **82.0**±0.4 | 79.7±1.8 | 17.4±8.5 | 17.8±19.3 | 59.1 | 53.1 | 11.4±6.3 | 57.5±8.3 | 79.5±1.0 |
| halfcheetah-med-expert | **105.0**±0.2 | 97.9±4.4 | 40.6±6.6 | 63.3±38.0 | 53.4 | 64.7 | 67.6±13.2 | 27.1±3.9 | 96.1±9.5 |
| hopper-med-expert | **112.7**±0.0 | 112.2±0.2 | 95.4±23.5 | 23.7±6.0 | 96.3 | 110.9 | 89.6±27.6 | 111.4±1.2 | 90.6±43.3 |
| walker2d-med-expert | **108.4**±1.3 | 101.1±9.3 | 14.8±1.4 | 44.6±12.9 | 40.1 | 57.5 | 12.0±5.8 | 68.1±13.1 | 103.6±4.6 |
| halfcheetah-expert | 107.6±0.9 | 105.7±1.9 | 104.0±2.0 | - | **108.2** | - | 105.2±1.7 | 82.4±7.4 | 106.8±3.0 |
| hopper-expert | 112.4±0.1 | 112.2±0.2 | 109.1±3.9 | - | 110.3 | - | 111.5±1.3 | 111.2±2.1 | 112.3±0.2 |
| walker2d-expert | **108.6**±1.5 | 105.7±2.7 | 88.4±3.7 | - | 106.1 | - | 56.0±24.9 | 103.8±7.6 | 79.9±32.4 |
| Total Score | **1020.7** | 979.3 | 618.6 | - | 802.6 | - | 595.3 | 764.3 | 974.6 |