# OpenReview forum: "Double Check Your State Before Trusting It: Confidence-Aware Bidirectional Offline Model-Based Imagination"
_NeurIPS.cc/2022/Conference — NeurIPS 2022 Accept_

### Official Review · Reviewer_ytiJ · 2022-07-11

**Rating:** 7
**Confidence:** 5
**Soundness:** 4 excellent
**Presentation:** 4 excellent
**Contribution:** 3 good

**Summary:**

The paper proposes a method of augmenting an offline dataset for use with model-free offline methods by generating synthetic rollouts that are “double-checked” - i.e., validated by a bidirectional transition model. The paper powerfully demonstrates the principle on a toy environment, and the proposed enhancement leads to a significant performance gain for BCQ on Adroit but a more minor gain for TD3+BC on MuJoCo.


**Questions:**

- Could standard offline model-based algorithms be used with the improved rollouts? Would there be a performance gain by only filtering out inaccurate forward rollouts in an algorithm like MOPO?
- Are there some states that are consistently discarded due to model inaccuracies? What proportion of states would be rejected with a 1-step rollout and double check?
- How was the ratio of real data to imagined trajectories tuned?


**Limitations:**

The authors discuss the limitations of their work well. Two points are:

- The authors address the limited gain on the MuJoCo-based environments, but could demonstrate the method is still useful by selecting a different algorithm to augment.
- [1, 2] show how inaccurate rollouts that are well penalized can still be useful for model-based training, and this point would be good to discuss.

[1] MOPO: Model-based Offline Policy Optimization Tianhe Yu, Garrett Thomas, Lantao Yu, Stefano Ermon, James Zou, Sergey Levine, Chelsea Finn, Tengyu Ma

[2] Revisiting Design Choices in Offline Model-Based Reinforcement Learning. Cong Lu, Philip J. Ball, Jack Parker-Holder, Michael A. Osborne, Stephen J. Roberts.


**Strengths And Weaknesses:**

Strengths:
- Clearly articulated method for validating the trajectories generated by model rollouts, strong illustrative toy example
- Enables model rollouts to be integrated with standard offline model-free RL algorithms with strong performance on the Adroit environments
- Extensive ablations against different data generation procedures and action-augmentation schemes

Weaknesses:
- Limited gain on the MuJoCo environments which authors attribute to the BC term in TD3+BC, would it be possible to see a similar strong gain with an alternative such as CQL?

Minor:
- Line 116: perhaps too strong a claim, certain offline model-free algorithms exhibit strong performance (IQL, TD3+BC) without imagined rollouts
- Line 146: training for a fixed number of epochs could lead to over/underfitting, is there a validation dataset?
- The number of seeds should be listed in table captions.
- Evaluation protocol for determining offline performance should be described (e.g. policy taken at the end of training?)

---

> ### Author Response · Authors · 2022-08-02
> **Author Response to Reviewer ytiJ (part 1/2)**
>
> Thanks for your detailed and valuable comments. We conduct additional experiments to clarify the raised questions and concerns. If you have any further questions or comments, we will be happy to have further discussions.
>
> **Q1: Can CQL have better performance gain with CABI compared with TD3_BC?**
>
> **A1:** Thanks for the insightful comment. The performance gain upon TD3_BC or IQL is limited since the generated samples much resemble the original samples in the static dataset, which makes it hard for TD3_BC (with behavior cloning term) or IQL (that learns without querying OOD samples) to exhibit significant performance gain. We deem that it is interesting to investigate whether CQL can have better performance gain with our CABI compared with TD3_BC. Due to the time limit, we can only run CQL+CABI over 4 different random seeds without tuning real data ratio $\eta$. To be specific, we use real data ratio $\eta=0.3$ for *random* datasets, $\eta=0.7$ for *medium* and *medium-replay* datasets, and a comparatively large $\eta=0.9$ for *medium-expert* and *expert* datasets (since they are of good quality). The forward horizon and backward horizon for rollout are set to be 3 for all of the datasets, which is consistent with the experimental setup for TD3_BC+CABI. We keep the original hyperparameters of CQL fixed. We summarize the experimental results in Table 1, where we observe that CQL does get large performance gain with the aid of CABI on all of the datasets. These altogether illustrate the effectiveness and benefits of our data augmentation method for offline learning.
>
> | Task Name | CQL | CQL+CABI |
> | ---- | :---: | :---: |
> | halfcheetah-random | 21.7$\pm$0.9 | **30.2$\pm$1.4** |
> | hopper-random | 10.7$\pm$0.1 | **13.5$\pm$3.5** |
> | walker2d-random | 2.7$\pm$1.2 | **7.3$\pm$2.3** |
> | halfcheetah-medium | 37.2$\pm$0.3 | **42.4$\pm$0.7** |
> | hopper-medium | 44.2$\pm$10.8 | **57.3$\pm$12.9** |
> | walker2d-medium | 57.5$\pm$8.3 | **62.7$\pm$6.4** |
> | halfcheetah-medium-replay | 41.9$\pm$1.1 | **44.6$\pm$0.4** |
> | hopper-medium-replay | 28.6$\pm$0.9 | **34.8$\pm$2.4** |
> | walker2d-medium-replay | 15.8$\pm$2.6 | **21.4$\pm$3.1** |
> | halfcheetah-medium-expert | 27.1$\pm$3.9 | **35.3$\pm$4.8** |
> | hopper-medium-expert | 111.4$\pm$1.2 | **112.0$\pm$0.4** |
> | walker2d-medium-expert | 68.1$\pm$13.1 | **107.5$\pm$1.0** |
> | halfcheetah-expert | 82.4$\pm$7.4 | **99.2$\pm$4.5** |
> | hopper-expert | 111.2$\pm$2.1 | **112.0$\pm$0.2** |
> | walker2d-expert | 103.8$\pm$7.6 | **110.2$\pm$0.9** |
> | Total score | 764.3 | **890.4** |
>
> Table 1. Normalized average score comparison on MuJoCo "-v0" datasets. The results of CQL+CABI are averaged over 4 different random seeds.
>
> **Q2: Could standard offline model-based algorithms be used with the improved rollouts? Can MOPO get performance gain by filtering forward imaginations with double check?**
>
> **A2:** Standard offline model-based algorithms like MOPO can also benefit from the improved rollouts. Model-based methods suffer from model inaccuracy and can generate poor synthetic transitions, especially when the rollout length is large since the model error will compound. With a double check mechanism, we can improve the quality of the imagined samples, and can benefit offline model-based algorithms. Empirically, we filter imaginations in MOPO, which we refer to as MOPO (filtering), and evaluate its performance on 3 random datasets and 3 medium-replay datasets from MuJoCo "-v0" datasets. For MOPO (filtering), we train bidirectional dynamics models (no rollout policies are trained) and we only adopt the backward dynamics model for transition filtering, i.e., we conduct double check on the forward imagination in MOPO. Note that MOPO adopts different rollout lengths for different datasets (see https://github.com/tianheyu927/mopo/tree/master/examples/config/d4rl). The rollout length for *walker2d-random*, *walker2d-medium-replay* are set to be 1. The rollout length is 5 for the rest. Since one-step rollout can be generally accurate, we keep 80\% samples for *walker2d-random* and *walker2d-medium-replay*. We keep 20\% samples for the rest of the datasets.
>
> We summarize the results in Table 2, where we observe performance improvement on all of the datasets. That validates our claims above.
>
> | Task Name | MOPO | MOPO (filtering) |
> | ---- | :---: | :---: |
> | halfcheetah-random | 35.4 | **38.1$\pm$1.8** |
> | hopper-random | 11.7 | **22.2$\pm$12.9** |
> | walker2d-random | 13.6 | **14.9$\pm$3.3** |
> | halfcheetah-medium-replay | 53.1 | **54.7$\pm$1.3** |
> | hopper-medium-replay | 67.5 | **72.2$\pm$17.0** |
> | walker2d-medium-replay | 39.0 | **41.3$\pm$1.1** |
>
> Table 2. Normalized average score on some D4RL MuJoCo "-v0" datasets. MOPO (filtering) refers to the variant of MOPO where we train bidirectional dynamics models in MOPO while only use the backward model for transition filtering. The results are averaged over 4 different random seeds.

---

> > ### Author Response · Authors · 2022-08-02
> > **Author Response to Reviewer ytiJ (part 2/2)**
> >
> > **Q3: Are there some states that are consistently discarded due to model inaccuracies? What proportion of states would be rejected with a 1-step rollout and double check?**
> >
> > **A3:** Thanks for the interesting questions. In Figure 4 of the main text, we show that either forward imagination and backward imagination is unreliable, as many invalid states are generated. While CABI can consistently reject those states. Therefore, there are some imagined states that are consistently discarded due to the disagreement between forward model and backward model in CABI. In our experiments, we keep $k$ unchanged. That is, even for 1-step rollout, the double check mechanism will reject 80\% samples. We want to note here that we only adopt 1-step rollout for *pen-human*, *pen-cloned*, *pen-expert* and *hammer-expert*. Because the model disagreement is large for larger horizons on those datasets (fitting these complex high-dimensional datasets can be difficult, please see Table 5 in the appendix) and we find reject 80\% samples for them is better. For simple tasks like MuJoCo, one ought not to reject 80\% imagined transitions when 1-step rollout is adopted (we keep 80\% transitions in MOPO (filtering) for 1-step rollout).
> >
> > **Q4: How was the ratio of real data to imagined trajectories tuned?**
> >
> > **A4:** The real data ratio $\eta$ is a vital hyperparameter for CABI. The real data ratio is tuned by using grid search in practice. To be specific, after training bidirectional dynamics models and rollout policies, we get synthetic offline dataset $\mathcal{D}\_m$. Suppose the batch size is $M$. Then we sample $\eta M$ samples from the raw static offline dataset and $(1-\eta)M$ transitions from the augmented dataset $\mathcal{D}\_m$ for the training of any model-free offline RL algorithm (1M steps). We tune the value of $\eta$ and then evaluate the performance of the model-free offline RL algorithms (1M steps) to pick the best possible real data ratio $\eta$. Note that one does not need to sweep across all $\eta$. The real data ratio $\eta$ is highly related to the quality of the dataset, i.e., for high-quality dataset, a large $\eta$ is expected while for datasets of poor quality, a small $\eta$ is better. This can help decrease the number of trials to find the best $\eta$.
> >
> > **Q5: Minor issues**
> >
> > **A5:** Thanks for your helpful comment. For line 116, we will modify our claim into "*Many* model-free offline RL algorithms suffer from poor generalization as they are trained on a fixed dataset with limited samples", which we believe is more suitable. For line 146, we use a validation set of 1000 transitions (see Appendix A line 38-39).
> >
> > The number of seeds are reported in the main text (see line 241 and line 290). We appreciate the advice from the reviewer and will add the number of seeds in the table captions. Following TD3_BC, our results are averaged over the final 10 evaluations.
> >
> > **Q6: Discussion on [1,2] that show how inaccurate rollouts that are well penalized can still be useful for model-based training**
> >
> > **A6:** We thank the reviewer for recommending these papers. MOPO is a model-based offline RL algorithm, and [2] investigates the common design choice (like penalty, rollout length, etc.). The main differences between [1,2] and CABI are in: (1) both [1,2] leverage the learned dynamics model for policy optimization. CABI, however, merely utilizes the learned dynamics models for reliable synthetic data generation. Then, a model-free offline RL algorithm is used to train on them; (2) CABI is proposed to increase the generalization capability of the model-free offline RL algorithms while MOPO is not; (3) CABI involves transition rejection while [1,2] accepts all imagined transitions.
> >
> > We find it may be a good direction to combine the idea of [2] in CABI, which we leave as future work. We will add the discussions in the appendix, and cite the papers recommended by the reviewer, which we think are of great value to our paper.
> >
> > [1] MOPO: Model-based Offline Policy Optimization Tianhe Yu, Garrett Thomas, Lantao Yu, Stefano Ermon, James Zou, Sergey Levine, Chelsea Finn, Tengyu Ma
> >
> > [2] Revisiting Design Choices in Offline Model-Based Reinforcement Learning. Cong Lu, Philip J. Ball, Jack Parker-Holder, Michael A. Osborne, Stephen J. Roberts.

---

> > > ### Comment · Reviewer_ytiJ · 2022-08-04
> > > **Thanks!**
> > >
> > > Thank you for the additional interesting results - even more evidence that the algorithm is extremely general!

---

> > > > ### Author Response · Authors · 2022-08-04
> > > > **Thanks for raising the score!**
> > > >
> > > > We appreciate the reviewer for raising the score to 7! Thanks for the valuable comments and suggestions!

---

### Official Review · Reviewer_rJ1S · 2022-07-11

**Rating:** 6
**Confidence:** 4
**Soundness:** 3 good
**Presentation:** 3 good
**Contribution:** 3 good

**Summary:**

This paper presents a new offline model-based RL method that trains a bidirectional model for model rollouts and filters out model-generated states given by the forward model that cannot recover the previous real state from the backward model. By doing so, the method can effectively get rid of inaccurate model-generated states and mitigate the distributional shift problem in offline model-based RL. Through extensive experiments, the method is able to outperform prior offline RL methods on various tasks in D4RL.

**Questions:**

See cons in the above section.

**Limitations:**

Yes

**Strengths And Weaknesses:**

**Pros:**

1. The method appears to be novel and addresses an important problem in offline RL, i.e. how to incorporate augmented data without introducing much distributional shift.

2. The results of the method on challenging domains such as Adroit are clearly better than the prior methods listed.

3. The paper is clearly written and easy to understand.

**Cons:**

1. The bidirectional models seem to assume that the action is reversible, which I think might not always hold in all MDPs. While the empirical results seem to suggest that incorporating such rollouts double-checked by the bidirectional models is helpful, there are definitely cases where the previous state is not possible to be inferred from the next state and the action, e.g. robotic manipulation situations that are contact-rich. I wonder if the authors can show the methods can also work in those situations (both empirically and theoretically).

2. The empirical evaluations are not that thorough. I believe one important baseline that is missing is S4RL, which also handles data augmentation in offline RL and achieves impressive results in adroit (even better results than the proposed method after skimming through their results table). The authors also didn't compare to MOReL, which is another important model-based offline RL method. The authors should also include evaluations in other domains such as antmaze and kitchen from D4RL.

3. It is a bit strange that the numbers of the baselines in the paper are different from/worse than ones in the original papers, e.g. CQL etc. I think it is important to show the original numbers to ensure fair comparisons.

---

> ### Author Response · Authors · 2022-08-02
> **Author Response to Reviewer rJ1S (part 1/2)**
>
> Thanks for your inspiring and thoughtful comments. We provide clarification to your questions and concerns as below. If you have any further questions or comments, please let us know, and we will be happy to have further discussions.
>
> **Q1: Can CABI still work in situations where states or actions are irreversible?**
>
> **A1:** Thanks for the interesting question. We acknowledge that there are cases where states or actions are irreversible, i.e., previous states cannot be inferred based on the current state (for example, the current state is an initial state and its previous state does not exist or is invalid). We argue that mere backward imagination may suffer from such situation. While our method, CABI, can mitigate this concern with the aid of *double check*. When a state $s\_t$ is irreversible, the disagreement between the forward model and backward model will be large. Then the generated (backward) synthetic transition from $s\_t$ will not be added into the model buffer. One can also see such evidence in our toy example (section 4.1 and Figure 4). In our toy RiskWorld datasets, there exist some states that are irreversible, e.g., the states that lie in the boundary. There also exists a danger zone in the RiskWorld task, and it is invalid to have samples in this zone. We can see from Figure 4(c) that backward model generates many invalid transitions that lie out of the support of the dataset or lie in the dangerous zone. However, CABI guarantees a good and reliable data generation where no invalid states are included.
>
> It is hard to theoretically show that CABI can work when states or actions are irreversible with the involvement of deep neural nets and the double check scheme. We empirically show in Table 2 of the main text that CABI can work in these situations, or at least can bring some performance gains, outperforming mere forward imagination, mere backward imagination, or mere bidirectional imagination without double check. The Adroit tasks are kind of similar to robotic manipulation tasks, and we believe experiments on them can explicitly show the effectiveness of our CABI.
>
> **Q2: missing comparison to S4RL and MOReL and CABI ought to be evaluated on other domains.**
>
> **A2:** We thank the reviewer for the kind advice and will add experiments of CABI on antmaze domain in the revision. We want to clarify here that S4RL [1] may not be a suitable baseline. In S4RL, the data augmentation is done by perturbing the states with various tricks. One important difference between CABI and S4RL lies in the fact that S4RL utilizes the augmented states for encouraging smoothness and consistency, and it is realized by modifying the standard bellman error (average Q over augmented states as below, please also see Equation (4) in S4RL).
> $$
> \min\_Q \mathbb{E}\_{s\_t,a\_t\sim\mathcal{D}}\left[r_t + \gamma \frac{1}{i}\sum\_i Q(\mathcal{T}\_i(\tilde{s}\_{t+1}|s\_{t+1}),a\_{t+1}) - \frac{1}{i}\sum\_i Q(\mathcal{T}\_i(\tilde{s}\_{t}|s\_{t}),a\_{t})\right]^2,
> $$
> where $\mathcal{T}(\tilde{s}\_{t}|s\_{t})$ is the data augmentation transformation defined in S4RL.
>
> CABI, instead, aims at generating *reliable* synthetic transitions with double check, and we make ***no modification*** to the base model-free offline RL algorithms. Also, S4RL only augments states while CABI generates *full* transitions, i.e., $(s\_t, a\_t, r\_t, s\_{t+1})$. The key idea and the way of utilizing the augmented samples are different for S4RL and CABI. We provide a performance comparison of BCQ+S4RL against vanilla BCQ and BCQ+CABI in Table 1. Since we cannot find the official codebase for S4RL, we implement it by following the instructions and hyperparameter setup in the original paper. We only implement S4RL(N) due to the time limit. We run BCQ+S4RL(N) on 8 datasets from Adroit over 4 different random seeds. We find that S4RL helps BCQ improve performance on some *human* datasets and *expert* datasets. While it slightly corrupts the performance of BCQ on *pen-expert*, *door-expert*, *relocate-human* datasets. It is dangerous to add noise to high-quality data such as expert data or human demonstrations since a small perturbation may make the augmented state out-of-distribution (OOD).
>
> | Task Name | BCQ | BCQ+CABI | BCQ+S4RL($\mathcal{N}$) |
> | ---- | :---: | :---: | :---: |
> | pen-human | 68.9 | 75.1$\pm$1.5 | 70.6$\pm$3.3 |
> | door-human | 0.0 | 1.7$\pm$0.1 | 0.1$\pm$0.0 |
> | relocate-human | 0.5 | 0.1$\pm$0.1 | 0.0$\pm$0.1 |
> | hammer-human | 0.5 | 3.1$\pm$2.2 | 1.9$\pm$2.7 |
> | pen-expert | 114.9 | 127.6$\pm$2.0 | 106.4$\pm$4.3 |
> | door-expert | 99.0 | 105.3$\pm$0.5 | 95.4$\pm$8.8 |
> | relocate-expert | 41.6 | 105.9$\pm$1.0 | 53.9$\pm$7.7 |
> | hammer-expert | 107.2 | 128.9$\pm$0.9 | 116$\pm$10.4 |
> | Total score | 432.6 | 547.7 | 444.3 |
>
> Table 1. Normalized average score comparison of BCQ, BCQ+CABI against BCQ+S4RL(N). The experiments are conducted on Adroit "-v0" datasets. The results of BCQ+S4RL(N) are averaged over 4 random seeds.

---

> > ### Author Response · Authors · 2022-08-02
> > **Author Response to Reviewer rJ1S (part 2/2)**
> >
> > **A2 Continued**
> >
> > For additional comparison to MOReL, we conduct experiments on 12 Adroit datasets over 4 random seeds using the reproduced code from github (https://github.com/SwapnilPande/MOReL). We directly take the results of MOReL reported in [2] on MuJoCo datasets due to the limited time and compute resources. The comparison of our methods against MOReL on Adroit datasets and MuJoCo datasets are available in Table 2 and 3, respectively. The results in Table 2 show that MOReL exhibits good performance on 4 datasets, while BCQ+CABI significantly outperforms MOReL on 8 out of 12 datasets. MOReL fails on *expert* datasets, while BCQ+CABI does not. The results in Table 3 show that MOReL exhibits good performance on some *random* and *medium-replay* datasets. While on other datasets, especially those contain expert demonstrations, our CQL+CABI and TD3_BC+CABI outperform MOReL.
> >
> > | Task Name | BCQ | BCQ+CABI | MOReL |
> > | ---- | :---: | :---: | :---: |
> > | pen-human | 68.9 | **75.1$\pm$1.5** | -3.2$\pm$0.1 |
> > | door-human | 0.0 | 1.7$\pm$0.1 | **2.2$\pm$0.2** |
> > | relocate-human | 0.5 | 0.1$\pm$0.1 | **2.3$\pm$1.0** |
> > | hammer-human | 0.5 | **3.1$\pm$2.2** | 2.7$\pm$0.5 |
> > | pen-cloned | 44.0 | **54.7$\pm$2.0** | -0.2$\pm$2.2 |
> > | door-cloned | 0.0 | 0.5$\pm$0.2 | **2.3$\pm$0.3** |
> > | relocate-cloned | -0.3 | -0.2$\pm$0.0 | **0.4$\pm$0.3** |
> > | hammer-cloned | 0.4 | **4.3$\pm$1.6** | 2.3$\pm$0.1 |
> > | pen-expert | 114.9 | **127.6$\pm$2.0** | -2.7$\pm$0.4 |
> > | door-expert | 99.0 | **105.3$\pm$0.5** | 2.4$\pm$0.9 |
> > | relocate-expert | 41.6 | **105.9$\pm$1.0** | 0.2$\pm$0.1 |
> > | hammer-expert | 107.2 | **128.9$\pm$0.9** | 2.9$\pm$0.1 |
> > | Total score | 476.7 | **607.0** | 11.6 |
> >
> > Table 2. Normalized average score comparison of BCQ, BCQ+CABI against MOReL. The results of MOReL are averaged over 4 random seeds.
> >
> > | Task Name | CQL | CQL+CABI | TD3_BC | TD3_BC+CABI | MOReL |
> > | ---- | :---: | :---: | :---: | :---: | :---: |
> > | halfcheetah-random | 21.7$\pm$0.9 | 30.2$\pm$1.4 | 10.2$\pm$1.3 | 15.1$\pm$0.4 | 25.6 |
> > | hopper-random | 10.7$\pm$0.1 | 13.5$\pm$3.5 | 11.0$\pm$0.1 | 11.9$\pm$0.1 | 53.6 |
> > | walker2d-random | 2.7$\pm$1.2 | 7.3$\pm$2.3 | 1.4$\pm$0.6 | 6.4$\pm$1.5 | 37.3 |
> > | halfcheetah-medium | 37.2$\pm$0.3 | 42.4$\pm$0.7 | 42.8$\pm$0.3 | 45.1$\pm$0.1 | 42.1 |
> > | hopper-medium | 44.2$\pm$10.8 | 57.3$\pm$12.9 | 99.5$\pm$1.0 | 100.4$\pm$0.3 | 95.4 |
> > | walker2d-medium | 57.5$\pm$8.3 | 62.7$\pm$6.4 | 79.7$\pm$1.8 | 82.0$\pm$0.4 | 77.8 |
> > | halfcheetah-medium-replay | 41.9$\pm$1.1 | 44.6$\pm$0.4 | 43.3$\pm$0.5 | 44.4$\pm$0.2 | 40.2 |
> > | hopper-medium-replay | 28.6$\pm$0.9 | 34.8$\pm$2.4 | 31.4$\pm$3.0 | 31.3$\pm$0.7 | 93.6 |
> > | walker2d-medium-replay | 15.8$\pm$2.6 | 21.4$\pm$3.1 | 25.2$\pm$5.1 | 29.4$\pm$1.3 | 49.8 |
> > | halfcheetah-medium-expert | 27.1$\pm$3.9 | 35.3$\pm$4.8 | 97.9$\pm$4.4 | 105.0$\pm$0.2 | 53.3 |
> > | hopper-medium-expert | 111.4$\pm$1.2 | 112.0$\pm$0.4 | 112.2$\pm$0.2 | 112.7$\pm$0.0 | 108.7 |
> > | walker2d-medium-expert | 68.1$\pm$13.1 | 107.5$\pm$1.0 | 101.1$\pm$9.3 | 108.4$\pm$1.3 | 95.6 |
> > | halfcheetah-expert | 82.4$\pm$7.4 | 99.2$\pm$4.5 | 105.7$\pm$1.9 | 107.6$\pm$0.9 | - |
> > | hopper-expert | 111.2$\pm$2.1 | 112.0$\pm$0.2 | 112.2$\pm$0.2 | 112.4$\pm$0.1 | - |
> > | walker2d-expert | 103.8$\pm$7.6 | 110.2$\pm$0.9 | 105.7$\pm$2.7 | 108.6$\pm$1.5 | - |
> > | Total score | 764.3 | 890.4 | 979.3 | 1020.7 | - |
> >
> > Table 3. Normalized average score comparison on MuJoCo "-v0" datasets. The results of CQL+CABI are averaged over 4 different random seeds. The results of TD3_BC+CABI are taken directly from the main text.
> >
> > [1] Sinha, et al. S4RL: Surprisingly Simple Self-Supervision for Offline Reinforcement Learning.
> >
> > [2] Kidambi, et al. MOReL: Model-Based Offline Reinforcement Learning.
> >
> > **Q3: Some baseline results are different from the original paper, e.g., CQL.**
> >
> > **A3:** We understand the concern. Actually, we *cannot reproduce* the results reported in the original CQL paper [1] and UWAC paper [2]. This may be because the dataset used by CQL is different from the current "-v0" datasets in D4RL (as mentioned in the author's github page, https://github.com/aviralkumar2907/CQL) In fact, the reported performance of CQL in the D4RL paper [3] (Table 2 in the appendix of D4RL paper) is different from the CQL original paper. We then take the results of CQL from TD3_BC paper directly. We have tried running the CQL using the official codebase on some "-v0" datasets, and the resulting performance is similar to the results reported in the TD3_BC paper. We then choose to trust those results and report them in our paper. For UWAC, we choose to re-run it on all datasets with its official codebase. We thus believe our evaluation is valid and reasonable.
> >
> > [1] Kumar, Aviral et al. Conservative Q-Learning for Offline Reinforcement Learning. NeurIPS.
> >
> > [2] Wu, Yue et al. Uncertainty Weighted Actor-Critic for Offline Reinforcement Learning. ICML.
> >
> > [3] Fu, Justin et al. D4RL: Datasets for Deep Data-Driven Reinforcement Learning. ArXiv.

---

> ### Author Response · Authors · 2022-08-08
> **Looking forward to your feedback**
>
> Dear reviewer rJ1S,
>
> We first would like to thank the reviewer's efforts and time in reviewing our work. We were wondering if our responses have resolved your concerns. We will be happy to have further discussions with the reviewer if there are still some remaining questions! More discussions and suggestions on further improving the paper are also always welcomed! We sincerely look forward to your kind reply!
>
> Best regards,
>
> The authors

---

### Official Review · Reviewer_iDjK · 2022-07-12

**Rating:** 6
**Confidence:** 5
**Soundness:** 3 good
**Presentation:** 3 good
**Contribution:** 2 fair

**Summary:**

The paper proposes a model-based data augmentation method (CABI) for offline RL, which uses the agreement of forward and backward models to assess the validity of transitions. It proceeds as follows:
1. Train forward and backward dynamics models, and forward and backward rollout policies based on CVAEs
2. Use models and rollout policies to generate transitions
3. Apply the backward model to transitions which were generated from the forward model, and vice-versa. Sort the transitions by the deviation of the predicted state from the original state, and keep only the transitions with relatively low deviation.

After generating these transitions, any model-free offline RL algorithm can be applied to the augmented dataset. Experiments on the D4RL benchmark demonstrate that the data augmentation is beneficial, improving over the base offline RL algorithm, in almost every case.

**Questions:**

* “We resort to sorting the transitions in a mini-batch by the state deviation from small to large and keep the top k% of them.” I’m guessing this is a typo and you meant the bottom k%? (Since you want small deviations)
* What is the exact formula used to compute ensemble variance? This should be included in the paper or appendix.

**Limitations:**

Yes, they mention that computational cost is a limitation of the work.
I agree that there is not significant risk of negative societal impact.

**Strengths And Weaknesses:**

Strengths
* The proposed double check is a novel idea, to my knowledge.
* The paper is clearly written.
* Experimental evaluation is thorough and demonstrates the performance benefits of CABI across a variety of datasets. The ablation study also shows the benefit of the double check compared to ensemble variance as an uncertainty quantifier.

Weaknesses
* Hyperparameter tuning: the appendix states that “The best ratio η strongly depends on the dataset and may need to be tuned manually.” One must also tune the rollout horizon.
* The improvement over the base offline RL algorithm, while consistent across tasks, is not large in most cases. This may be because the generated samples are very similar to the original samples (since only in-support actions are tried).

---

> ### Author Response · Authors · 2022-08-02
> **Author Response to Reviewer iDjK**
>
> We thank the reviewer for the insightful comments. We hope our responses below can address your concerns. If the reviewer has any further questions or comments, we are happy to have a discussion.
>
> **Q1: On hyperparameter tuning**
>
> **A1:** The real data ratio $\eta$ is a vital hyperparameter for CABI. This parameter is highly related to the *quality* of the static dataset, e.g., if the offline dataset is of high expert quality, then we expect a relatively large $\eta$; while if the logged dataset is of poor quality (e.g., random datasets), then we expect a relatively small $\eta$ since the real data itself is unreliable for training. In practice, one does need to decide the real data ratio $\eta$. As for the horizon, one can simply keep it unchanged on MuJoCo tasks. While for more complex and harder tasks (e.g., Adroit), one may need to find a suitable rollout horizon.
>
> **Q2: The improvement over the base offline RL algorithm is not large on some datasets.**
>
> **A2:** In our experiments, we observe remarkable performance improvement over the base BCQ algorithm on many Adroit datasets, while on MuJoCo domain, the performance improvement upon TD3$\\\_$BC is not that large. This is because, as the reviewer comments, the generated reliable transitions by CABI are still similar to the raw samples in the static offline dataset. Therefore, combining CABI with TD3$\\_$BC (with a behavior cloning term) and IQL (that learns without querying OOD samples) does not bring much performance improvement. Nevertheless, we observe that CABI is still beneficial for these methods. To better show the effectiveness of CABI in MuJoCo domain, we combine CABI with CQL and conduct extensive empirical experiments on 15 datasets over 4 different random seeds. We summarize the results in Table 1. We find that CABI brings larger performance improvement for CQL.
>
> | Task Name | CQL | CQL+CABI |
> | ---- | :---: | :---: |
> | halfcheetah-random | 21.7$\pm$0.9 | **30.2$\pm$1.4** |
> | hopper-random | 10.7$\pm$0.1 | **13.5$\pm$3.5** |
> | walker2d-random | 2.7$\pm$1.2 | **7.3$\pm$2.3** |
> | halfcheetah-medium | 37.2$\pm$0.3 | **42.4$\pm$0.7** |
> | hopper-medium | 44.2$\pm$10.8 | **57.3$\pm$12.9** |
> | walker2d-medium | 57.5$\pm$8.3 | **62.7$\pm$6.4** |
> | halfcheetah-medium-replay | 41.9$\pm$1.1 | **44.6$\pm$0.4** |
> | hopper-medium-replay | 28.6$\pm$0.9 | **34.8$\pm$2.4** |
> | walker2d-medium-replay | 15.8$\pm$2.6 | **21.4$\pm$3.1** |
> | halfcheetah-medium-expert | 27.1$\pm$3.9 | **35.3$\pm$4.8** |
> | hopper-medium-expert | 111.4$\pm$1.2 | **112.0$\pm$0.4** |
> | walker2d-medium-expert | 68.1$\pm$13.1 | **107.5$\pm$1.0** |
> | halfcheetah-expert | 82.4$\pm$7.4 | **99.2$\pm$4.5** |
> | hopper-expert | 111.2$\pm$2.1 | **112.0$\pm$0.2** |
> | walker2d-expert | 103.8$\pm$7.6 | **110.2$\pm$0.9** |
> | Total score | 764.3 | **890.4** |
>
> Table 1. Normalized average score comparison on MuJoCo "-v0" datasets. The results of CQL+CABI are averaged over 4 different random seeds.
>
> **Q3: A typo on top k\%**
>
> **A3:** We are sorry for the confusion here. In the main text, we state that "We resort to sorting the transitions in a mini-batch by the state deviation from small to large and keep the top k\% of them". Since we have already sorted the transitions by the deviation from small to large, the *top* 20\% (or the *first* 20\%) in this sorted batch will be transitions with the smallest deviation. To make our statement clearer, and to ease confusion here, we will modify this statement into "We resort to sorting the transitions in a mini-batch by the state deviation from small to large and keep k\% of them *that have the smallest deviation*". This modification will be revealed in the revision.
>
> **Q4: What is the exact formula used to compute ensemble variance?**
>
> **A4:** Thanks for the helpful comment. We take the ensemble rejection in forward dynamics model as an example. We train an ensemble of forward dynamics models, $f\_1(s\_{t+1}|s\_t), f\_2(s\_{t+1}|s\_t),\ldots,f\_N(s\_{t+1}|s\_t)$. For a given current state $s\_t$, we can then get an ensemble of next state $(\hat{s}\_{t+1}^1, \hat{s}\_{t+1}^2,\ldots,\hat{s}\_{t+1}^N)$. We then randomly pick one next state while recording the variance in the ensemble at the same time. We  then reject the generated next state if the variance in the ensemble is large. That is, we evaluate the variance of $(\hat{s}\_{t+1}^1, \hat{s}\_{t+1}^2,\ldots,\hat{s}\_{t+1}^N)$, i.e., $Var = \mathbb{E}\_{i=1}^N [(\hat{s}\_{t+1}^i - \mathbb{E}[\hat{s}\_{t+1}^i])^2]$. We sort the transitions in a batch by their calculated variance, and only trust the 20\% transitions that have the smallest *ensemble variance*. We will add this detail in the appendix.

---

### Meta-Review · Area_Chair_tjmx · 2022-08-20

**Recommendation:** Accept
**Confidence:** Certain

**Metareview:**

**Strengths**: The paper introduces a new and interesting idea of "double checking" with bi-directional models, and thoroughly evaluates the idea on a variety of offline RL datasets and through multiple ablations.

**Weaknesses**: The main weaknesses seem to be that (1) some of the performance improvements are small, and (2) like prior methods, the method is heavily reliant on tuning a hyperparameter based on the quality of the dataset. It also is somewhat strange that the paper uses v0 datasets from D4RL, since the more recent versions have fixed bugs in the datasets. Otherwise, the author response did a good job at discussing and addressing the other reviewer concerns.

The reviewers and AC agree that the strengths outweigh the weaknesses, and would make a valuable addition to NeurIPS.

**Award:**

No

---

### Decision · Program_Chairs · 2022-09-14

Accept